# WASSERSTEIN FLOW MATCHING: GENERATIVE MODELING OVER FAMILIES OF DISTRIBUTIONS

## ABSTRACT

Generative modeling typically concerns the transport of a single source distribution to a single target distribution by learning (i.e., regressing onto) simple probability flows. However, in modern data-driven fields such as computer graphics and single-cell genomics, samples (say, point-clouds) from datasets can themselves be viewed as distributions (as, say, discrete measures). In these settings, the standard generative modeling paradigm of flow matching would ignore the relevant geometry of the samples. To remedy this, we propose *Wasserstein flow matching* (WFM), which appropriately lifts flow matching onto families of distributions by appealing to the Riemannian nature of the Wasserstein geometry. Our algorithm leverages theoretical and computational advances in (entropic) optimal transport, as well as the attention mechanism in our neural network architecture. We present two novel algorithmic contributions. First, we demonstrate how to perform generative modeling over Gaussian distributions, where we generate representations of granular cell states from single-cell genomics data. Secondly, we show that WFM can learn flows between high-dimensional and variable sized point-clouds and synthesize cellular microenvironments from spatial transcriptomics datasets. Code is available at WassersteinFlowMatching.

## 1 INTRODUCTION

Today's abundance of data and scalability of training massive neural networks has made it possible to generate hyper-realistic images on the basis of training examples (OpenAI, 2022), as well as video and audio clips (Vyas et al., 2023; Xing et al., 2023), and, of course, text (Bubeck et al., 2023). All of these are instances of generative modeling: given access to finitely many samples from a distribution, devise a scheme which generates new samples from the same distribution. Generative modeling has also been revolutionary in the biomedical sciences, for drug design (Jumper et al., 2021) and single-cell genomics (Lopez et al., 2018). Nearly all frameworks exploit the notion that datasets (of, say, genomic profiles of cells, images, videos, or corpora of text documents) are instantiations of probability measures, and the task is to transform a point sampled from random noise to generate a data point that obeys the distribution of interest.

Among the zoo of available generative models, one approach noted for its flexibility and simplicity is Flow Matching (FM) (Albergo & Vanden-Eijnden, 2022; Lipman et al., 2022; Liu et al., 2022). For a fixed target probability measure, FM learns an implicitly defined vector field that can transform a source measure (e.g., the standard Gaussian) to the target. Unlike discrete time and probabilistic generative models (such as Denoising Diffusion Models by Song et al. (2020)), FM learns a deterministic, continuous normalizing flow by regressing onto a simple conditional probability flow. This approach, while originally designed for Euclidean domains, can be readily adopted to Riemannian geometries (Chen & Lipman, 2023). Riemannian flow matching is widely used for generating samples over geometries such as spheres, tori, translation/rotation groups, simplices, triangular meshes, mazes, and molecular positions and structures.

The *Wasserstein geometry*, a canonical geometry over distributions, does not easily fit into any of these existing frameworks and has not been successfully adapted for flow matching. This geometry is useful, for example, in computational graphics where collections of 3D shapes are represented as empirical distributions (point-clouds). Likewise, recent developments in single-cell genomics analysis have demonstrated that gene-expression profiles from groups of cells aggregated via their

| Method | Data type | Source | Target |
|---|---|---|---|
| FM over $\mathbb{R}^d$ | $x \in \mathbb{R}^d$ | $x \sim p_0$ | $y \sim p_1$ |
| FM over $\mathcal{M}$ | $x \in \mathcal{M}$ | $x \sim \mathfrak{p}_0$ | $y \sim \mathfrak{p}_1$ |
| FM over $\Delta_d$ | $\mu \in \mathcal{P}(\Delta_d)$ | $\mu \sim \mathfrak{p}_0$ | $\nu \sim \mathfrak{p}_1$ |
| Wasserstein FM | $\mu \in \mathcal{P}(\mathbb{R}^d)$ | $\mu \sim \mathfrak{p}_0$ | $\nu \sim \mathfrak{p}_1$ |
| $\rightarrow$ Gaussians | $\mathcal{N}(m, \Sigma)$ | $\mathcal{N}(m_\mu, \Sigma_\mu) \sim \mathfrak{p}_0$ | $\mathcal{N}(m_\nu, \Sigma_\nu) \sim \mathfrak{p}_1$ |
| $\rightarrow$ Point-Clouds | $\frac{1}{n}\sum_i \delta_{x_i}$ | $\frac{1}{m}\sum_i \delta_{x_i} \sim \mathfrak{p}_0$ | $\frac{1}{n}\sum_j \delta_{y_j} \sim \mathfrak{p}_1$ |

Figure 1: *Left:* Table contrasting FM methods over $\mathbb{R}^d$, general manifolds $\mathcal{M}$, categorical and Dirichlet distributions on the $d$-simplex $\Delta_d$, and finally, our approach, FM problems defined over $\mathcal{P}(\mathbb{R}^d)$. *Right:* WFM overview, which learns flows over distributions over distributions.

mean and covariance can capture cellular microenvironments or highlight fine-grain clusters (Haviv et al., 2024b; Persad et al., 2023). For both point-cloud and Gaussian settings, it is natural to search for a unified generative model that respects the underlying geometry of the data, namely, treating each sample as itself a probability distribution.

**Contributions.** We introduce *Wasserstein Flow Matching* (WFM), a principled extension of the FM framework lifted to the space of probability distributions. As illustrated in Figure 1, a single point in our source and target datasets is itself a distribution (e.g., a single discrete measure or a single Gaussian), and our aim is to learn vector fields acting on the space of probability distributions and match the optimal transport map, which is the geodesic in Wasserstein space. WFM is an instantiation of Riemannian FM (Chen & Lipman, 2023), where we train a neural model to learn a continuous normalizing flow (CNF) between distributions over distributions.

We demonstrate the effectiveness of our approach for generative modeling between distributions over Gaussian distributions and distributions over point-clouds. The former task is motivated by recent directions in single-cell and spatial transcriptomics (Haviv et al., 2024b; Persad et al., 2023), where we consider matching problems over the *Bures–Wasserstein space* (BW), the Gaussian sub-manifold of the Wasserstein space. In this case, we show that WFM can be further modified, resulting in the Bures–Wasserstein FM (BW-FM) algorithm. We validate BW-FM on a variety of Gaussian-based datasets, where we observe that samples generated by our algorithm are significantly more robust than naïve approaches which do not fully exploit the underlying geometry of the data. In turn, we present a generative model for cell states and niches from single-cell genomics data.

Point-cloud generation is made possible by two distinct, yet crucial, algorithmic primitives: (1) incorporating transformers in our neural network architecture (Vaswani, 2017; Lee et al., 2019), and (2) recent algorithmic advances in entropic optimal transport (Pooladian & Niles-Weed, 2021). Indeed, our WFM algorithm performs generative modeling in the Wasserstein space, where geodesics are given by pushforwards of optimal transport (OT) maps; see Section 2.3 for more information. Both the transformer architecture and entropic optimal transport are crucial to approximating the OT map between independent point-clouds. Indeed, the permutation equivariance of attention makes the transformer a natural basis for our model, inherently modeling the equivariance feature of the Wasserstein geometry while maintaining scalability in high-dimensions.

For datasets of 3D point-clouds with uniform sizes, the performance of WFM is comparable to other current generative models. However, due to their particular training paradigms (namely the voxelization of 3D spaces), contemporary approaches cannot scale to high-dimensional point-clouds and fail on datasets with variable sized examples. Conversely, WFM succeeds in the high-dimensional and inhomogeneous settings, unlocking generative modeling to new, previously uncharted domains such as synthesizing niches from spatial genomics data. The ability to model tissue biology in this generative manner could enhance our understanding of how environment is associated with cell state. In the context of many diseases, most notably cancer and its tumor-immune microenvironment, these insights are critical for developing novel therapeutics (Binnewies et al., 2018).

## 2 BACKGROUND AND RELATED WORK

We let $\mathcal{P}_2(\mathbb{R}^d)$ denote the set of probability distributions over $\mathbb{R}^d$ with finite second moment, and write $\mathcal{P}_{2,\text{ac}}(\mathbb{R}^d)$ to be those with densities. For a probability measure $\mu$ and (vector-valued) function

$f$, we interchangeably write $\int \|f(x)\|^2 \, \mathrm{d}\mu(x)$ and $\|f\|_{L^2(\mu)}^2$. Let $\mathcal{M}$ be a Riemannian manifold, with $\mathcal{P}(\mathcal{M})$ defining the space of probability measures over said manifold. For $x \in \mathcal{M}$, we write $\mathcal{T}_x\mathcal{M}$ to mean the tangent space of the manifold at $x$, and write the metric on the tangent space (at $x$) as $g(x)$. For $x_0 \in \mathcal{M}$ with initial velocity $v \in \mathcal{T}_{x_0}\mathcal{M}$, the terminal location of the resulting geodesic is expressed as the output of the exponential map $v \mapsto \exp_{x_0}(v) \in \mathcal{M}$. Similarly, for an initial point $x_0$ and terminal location $x_1$, the logarithmic map defines the tangent vector, denoted $x_1 \mapsto \log_{x_0}(x_1)$, such that $\exp_{x_0}(\log_{x_0}(x_1)) = x_1$. The set of symmetric matrices (resp. positive definite matrices) over $\mathbb{R}^d$ are denoted by $\mathbb{S}^d$ (resp. $\mathbb{S}^d_{++}$).

## 2.1 RIEMANNIAN FLOW MATCHING

We first briefly discuss the Riemannian flow matching (RFM) framework of Chen & Lipman (2023). Let $\mathfrak{p}_0$ be the source distribution and $\mathfrak{p}_1$ be the target distribution over a Riemannian manifold $\mathcal{M}$, and let $(\gamma_t)_{t \in [0,1]}$ be a curve of probability measures satisfying $\gamma_0 = \mathfrak{p}_0$ and $\gamma_1 = \mathfrak{p}_1$. Letting $(w_t)_{t \in [0,1]}$ denote a family of vector fields, we say that the pair $(\gamma_t, w_t)_{t \in [0,1]}$ satisfy the *continuity equation* with respect to the metric $g$, abbreviated to $(\gamma_t, w_t) \in \mathfrak{C}_g$ if

$$\partial_t \gamma_t + \nabla_g \cdot (\gamma_t w_t) = 0 \,, \tag{1}$$

where $\nabla_g \cdot$ is the Riemannian divergence operator.

The goal of RFM is to regress a parameterized vector field (e.g., a neural network), written $f_\theta(x, t) \in \mathcal{T}_x\mathcal{M}$ for $t \in [0,1]$, onto the family $w_t$ by minimizing

$$\min_\theta \int_0^1 \int \|f_\theta(z_t, t) - w_t(z_t)\|_{g(z_t)}^2 \, \mathrm{d}\gamma_t(z_t) \, \mathrm{d}t \,,$$

assuming access to a pair $(\gamma_t, w_t)_{t \in [0,1]}$ that satisfies (1), which is not possible in many scenarios. Borrowing insights from recent work (e.g., Albergo & Vanden-Eijnden (2022); Lipman et al. (2022); Liu et al. (2022)), the authors construct a simple vector field that satisfies the continuity equation, resulting in the tractable objective

$$\min_\theta \int_0^1 \iint \|f_\theta(x_t, t) - \dot{x}_t\|_{g(x_t)}^2 \, \mathrm{d}\mathfrak{p}_0(x) \, \mathrm{d}\mathfrak{p}_1(y) \, \mathrm{d}t \,, \tag{2}$$

where, for example, $x_t = \exp_x((1-t)\log_x(y)) \in \mathcal{M}$, and $\dot{x}_t \in \mathcal{T}_{x_t}\mathcal{M}$. For complete discussions and proofs, see Chen & Lipman (2023, Section 3.1). Once $f_\theta$ is appropriately fit using (2), we can generate new samples from $\mathfrak{p}_1$: start by sampling $X_0 \sim \mathfrak{p}_0$, then follow $\dot{X}_t = f_\theta(X_t, t)$ numerically by discretizing the dynamics given by the exponential map, resulting in $X_1 \sim \mathfrak{p}_1$. We emphasize that the dynamics are only simulated at inference time and not when training $f_\theta$, which is commonly known as a *simulation-free* training paradigm.

## 2.2 RELATED WORK

**Generative models for point-clouds.** Paralleling the progress in generative models for natural images, the field of point-cloud generation is rapidly expanding. Many different models have been used from this task, namely generative-adversarial-nets (Achlioptas et al., 2018), variational autoencoders (Gadelha et al., 2018), normalizing flows (Yang et al., 2019; Kim et al., 2020; Klokov et al., 2020), diffusion (Zhou et al., 2021; Cai et al., 2020) and even euclidean FM (Wu et al., 2023). Thus far, these approaches are limited to uniformly sized point-clouds in 2D & 3D, and fail on high-dimensional spaces which cannot be voxelized.

**Generative models over families of distributions.** Our work is not the first to instantiate Riemannian FM with a manifold of probability measures. Two notable works are Fisher FM (Davis et al., 2024) and Categorical FM (Cheng et al., 2024), which consider the FM algorithm with respect to the Fisher–Rao geometry Amari (2016); Nielsen (2020) over the $d$-dimensional simplex $\Delta_d$. The work of Stark et al. (2024) is similar in spirit, where they focus on the Dirichlet distribution for generation of discrete data. Another related work is that of Atanackovic et al. (2024), called Meta FM. Their approach requires pairs of distributions which are already coupled, with the goal of solving FM between a distribution over pairs. In contrast, we emphasize that our proposed Wasserstein FM applies between two separate *uncoupled* distributions over distributions.

**Generative models for single-cell genomics.** Deep learning based generative models have transformed single-cell genomics through various approaches. Variational auto-encoders (Lopez et al., 2018; Gayoso et al., 2022) have successfully addressed technical artifacts, integrating multi-moodal data and imputing missing features in scRNA-seq data. More recently, Transformer-based foundation models have been noted for their ability to integrate large atlases of data (Cui et al., 2024; Theodoris et al., 2023). Flow Matching has also emerged as a promising direction (Klein et al., 2024; Eyring et al., 2024) for learning both balanced and unbalanced OT maps between cell populations. While these prior FM applications focus on cell-to-cell mappings, our work introduces a new paradigm: generating entire point-clouds representing cellular populations. This is particularly relevant for spatial genomics, where cellular microenvironments are naturally represented as point-clouds. WFM enables synthesis of whole cellular neighborhoods, a novel approach in generative modeling for the single-cell field.

## 2.3 WASSERSTEIN GEOMETRY

The (squared) 2-*Wasserstein distance* between two probability measures $\mu, \nu \in \mathcal{P}_{2,\mathrm{ac}}(\mathbb{R}^d)$ is given by the non-convex optimization problem over vector-valued maps $T : \mathbb{R}^d \to \mathbb{R}^d$

$$W_2^2(\mu, \nu) \coloneqq \min_{T : T_\sharp \mu = \nu} \|\mathrm{id} - T\|_{L^2(\mu)}^2 \,, \tag{3}$$

where the pushforward constraint, written $T_\sharp \mu = \nu$, means that, for $X \sim \mu$, the image follows $T(X) \sim \nu$. The minimizer to (3) is called the *optimal transport (OT) map*, denoted $T_\star^{\mu \to \nu}$ (we abbreviate this to $T_\star$ when it is clear from context). The existence and uniqueness of the optimal transport map under the stated regularity conditions is due to Brenier (1991).

The *Wasserstein space* is the space of probability densities with finite second moment endowed with the Wasserstein distance; this space is known to be a metric space (Villani, 2009). Following the celebrated work of Otto (2001), the Wasserstein space can be formally (meaning, non-rigorously) viewed as a Riemannian manifold, whose properties we now describe in brief; see e.g., Ambrosio et al. (2008) for a rigorous treatment.

Following the definition by Ambrosio et al. (2008, Theorem 8.5.1), the tangent space at a point $\mu \in \mathcal{P}_{2,\mathrm{ac}}(\mathbb{R}^d)$[1] consists of all possible tangent vectors that emanate from $\mu$, written formally as

$$\mathcal{T}_\mu \mathcal{P}_{2,\mathrm{ac}}(\mathbb{R}^d) \coloneqq \overline{\{\lambda(T_\star^{\mu \to \nu} - \mathrm{id}) \; : \; \lambda > 0, \nu \in \mathcal{P}_2(\mathbb{R}^d)\}}^{L^2(\mu)} \,,$$

where the overline denotes the closure of the set (i.e., the set and its limit points) in $L^2(\mu)$, and the norm on the tangent space is also $L^2(\mu)$. The exponential and logarithmic maps read

$$v \mapsto \exp_\mu(v) \coloneqq (\mathrm{id} + v)_\sharp \mu \,, \quad \nu \mapsto \log_\mu(\nu) \coloneqq T_\star^{\mu \to \nu} - \mathrm{id} \,,$$

where id is the identity map. Consequently, the (constant-speed) geodesic, or *McCann interpolation*, between two measures $\mu$ and $\nu$ is given by the curve $(\mu_t)_{t \in [0,1]}$ where

$$\mu_t \coloneqq (T_t^{\mu \to \nu})_\sharp \mu \coloneqq ((1-t)\mathrm{id} + t T_\star^{\mu \to \nu})_\sharp \mu \equiv \exp_\mu((1-t)\log_\mu(\nu)) \,, \tag{4}$$

where the last expression writes the pushforward in terms of the exponential and logarithmic maps. Equivalently, at the level of the random variables, one can write $X_t = (1-t)X_0 + t T_\star^{\mu \to \nu}(X_0)$, where $X_0 \sim \mu$ and $X_t \sim \mu_t$ for any $t \in [0,1]$. Combined with $(v_t)_{t \in [0,1]}$ a suitable family of vector fields, the McCann interpolation satisfies the continuity equation (1) over $\mathbb{R}^d$, re-written as

$$\partial_t \mu_t + \nabla \cdot (\mu_t v_t) = 0 \,, \quad \text{s.t.} \quad \mu_0 = \mu \,, \mu_1 = \nu \,, \tag{5}$$

where the divergence operator is the usual Euclidean one over $\mathbb{R}^d$, thus we write $(\mu_t, v_t) \in \mathfrak{C}$. The link between the constant speed geodesics and the 2-Wasserstein distance can be viewed from the celebrated Benamou–Brenier formulation of optimal transport (Benamou & Brenier, 2000):

$$W_2^2(\mu, \nu) = \inf_{(\mu_t, v_t) \in \mathfrak{C}} \int_0^1 \|v_t\|_{L^2(\mu_t)}^2 \, \mathrm{d}t \,. \tag{6}$$

---

[1]For Gaussians, $\mu$ is naturally absolutely continuous. For point-clouds, we interpret them as empirical samples drawn from underlying continuous shapes (e.g., a car's surface).

The optimal curve of measures is given by the constant-speed geodesics described above, and the optimal velocity field is given by

$$v_t = (T_\star^{\mu \to \nu} - \mathrm{id}) \circ (T_t^{\mu \to \nu})^{-1} \,. \tag{7}$$

The vector field (7) should be interpreted as the time-derivative of the McCann interpolation:

$$\dot{X}_t = (T_\star^{\mu \to \nu} - \mathrm{id})(X_0) = (T_\star^{\mu \to \nu} - \mathrm{id}) \circ (T_t^{\mu \to \nu})^{-1}(X_t) \,, \qquad X_0 \sim \mu \,.$$

### 2.3.1 Bures–Wasserstein (BW) space

A known special case of the Wasserstein space is the *Bures–Wasserstein* space, which consists of the submanifold of non-degenerate Gaussians parameterized by means and covariances $\{(m, \Sigma) : m \in \mathbb{R}^d, \Sigma \in \mathbb{S}_{++}^d\}$, endowed with the Wasserstein metric. We provide a brief exposition on the geometry of the Bures–Wasserstein space and refer the interested reader to Lambert et al. (2022) for detailed calculations and explanations, as we follow their notation conventions.

The OT map between $\mu = \mathcal{N}(m_\mu, \Sigma_\mu)$ and $\nu = \mathcal{N}(m_\nu, \Sigma_\nu)$ has a closed-form (Gelbrich, 1990):

$$T_\star(x) := m_\nu + C^{\mu \to \nu}(x - m_\mu) := b + \Sigma_\mu^{-\frac{1}{2}} (\Sigma_\mu^{\frac{1}{2}} \Sigma_\nu \Sigma_\mu^{\frac{1}{2}})^{\frac{1}{2}} \Sigma_\mu^{-\frac{1}{2}} (x - m_\mu) \,.$$

As this map is affine, it is clear that the McCann interpolation between two Gaussians is always Gaussian (indeed, Gaussians undergoing affine transformations remain Gaussian). More generally, we have the succinct representation of the tangent space at a point in the Bures–Wasserstein space

$$\mathcal{T}_\mu \mathrm{BW}(\mathbb{R}^d) := \{a + S(\mathrm{id} - m_\mu) \,:\, a \in \mathbb{R}^d, S \in \mathbb{S}^d\} \,,$$

and the exponential and logarithmic maps between two non-degenerate Gaussians are

$$(a, S) \mapsto \exp_\mu((a, S)) := \mathcal{N}(m_\mu + a, (S + I)\Sigma_\mu(S + I)) \,,$$

$$\nu \mapsto \log_\mu(\nu) := (m_\nu - m_\mu, \Sigma_\mu^{-\frac{1}{2}} (\Sigma_\mu^{\frac{1}{2}} \Sigma_\nu \Sigma_\mu^{\frac{1}{2}})^{\frac{1}{2}} \Sigma_\mu^{-\frac{1}{2}} - I) \,,$$

where the exponential map requires $S \succ -I$. We also mention that the norm on the tangent space at $\mu$ in the Bures–Wasserstein space can be written as

$$\|(a, S)\|_{\mathrm{BW}(\mu)}^2 := \|a - m_\mu\|^2 + \mathrm{Tr}(S^2 \Sigma_\mu)$$

With the above, it is easy to compute the closed-form solutions for the mean and covariance of the McCann interpolation $\mu_t = (T_t)_\sharp \mu = \mathcal{N}(m_t, \Sigma_t)$, given by

$$m_t := (1 - t)a + tb \,, \quad \Sigma_t := T_t A T_t := ((1 - t)I + tC^{A \to B})A((1 - t)I + tC^{A \to B}) \,. \tag{8}$$

We can relate the Euclidean and Riemannian time-derivatives of $\Sigma_t$ through the following manipulation (the latter of which respects the exponential and logarithmic maps above):

$$\dot{\Sigma}_t^{\mathrm{E}} = \dot{T}_t A T_t + T_t A \dot{T}_t = \dot{T}_t (T_t)^{-1} T_t A T_t + T_t A T_t (T_t)^{-1} \dot{T}_t = \dot{\Sigma}_t^{\mathrm{BW}} \Sigma_t + \Sigma_t \dot{\Sigma}_t^{\mathrm{BW}} \,.$$

To this end, we can draw parallels to (7) by writing

$$\dot{m}_t = b - a \,, \quad \dot{\Sigma}_t^{\mathrm{BW}} = (C^{A \to B} - I)((1 - t)I + tC^{A \to B})^{-1} \,. \tag{9}$$

## 3 Flow matching over the Wasserstein space

### 3.1 Training

Let $\mathfrak{p}_0$ and $\mathfrak{p}_1$ denote probability measures over the Wasserstein space.[2] Our goal is to learn a vector field that transports the family of measures $\mathfrak{p}_0$ to the family $\mathfrak{p}_1$. WFM learns to map source to target by regressing onto Wasserstein geodesics between samples $\mu \sim \mathfrak{p}_0$ and $\nu \sim \mathfrak{p}_1$, rather than learning the OT map between $\mathfrak{p}_0$ and $\mathfrak{p}_1$. To accomplish this, we pass in the McCann interpolation

---

[2]This implies that $\mu \sim \mathfrak{p}_0$ is itself a distribution (e.g., a Gaussian or a point-cloud), *not* a random variable.

**Algorithm 1:** Wasserstein FM Training

**Require:** base $\mathfrak{p}_0 \in \mathcal{P}(\mathcal{P}(\mathbb{R}^d))$, target
$\qquad\qquad \mathfrak{p}_1 \in \mathcal{P}(\mathcal{P}(\mathbb{R}^d))$, $\texttt{geo} \in \{\mathrm{BW}, \mathrm{PC}\}$
**Init:** Parameters $\theta$ of $f_\theta^{\texttt{geo}}$
**while** *not converged* **do**
$\quad$ Sample time $t \sim \mathcal{U}(0,1)$
$\quad$ Sample source measure $\mu \sim \mathfrak{p}_0$
$\quad$ Sample target measure $\nu \sim \mathfrak{p}_1$
$\quad$ **if** $\texttt{geo}$ *is BW* **then**
$\quad\quad \mu_t \leftarrow (m_t, \Sigma_t)$ via (8)
$\quad\quad v_t \leftarrow (\dot{m}_t, \dot{\Sigma}_t^{\mathrm{BW}})$ via (9)
$\quad$ **else**
$\quad\quad \mu_t \leftarrow$ Approximate via (4) using $\hat{T}^{\mu \to \nu}$
$\quad\quad v_t \leftarrow$ Approximate via (7)
$\quad \ell(\theta) \leftarrow \|f_\theta^{\texttt{geo}}(\mu_t, t) - v_t\|_{L^2(\mu_t)}^2$
$\quad \theta \leftarrow \texttt{optimizer\_step}(\theta, \ell(\theta), \nabla_\theta \ell(\theta))$

**Algorithm 2:** $\mathrm{BW}(\mathbb{R}^d)$ generation

**Data:** Trained $f_\theta^{\mathrm{BW}}$, step size $h = 1/N$
**Init:** $\mathcal{N}(m_0, \Sigma_0) \sim \mathfrak{p}_0$
**for** $k = 0, \ldots, N-1$ **do**
$\quad (s_k, S_k) \leftarrow f_\theta^{\mathrm{BW}}((m_{kh}, \Sigma_{kh}), kh)$
$\quad m_{(k+1)h} \leftarrow m_k + h s_k$
$\quad U_k \leftarrow (I + h S_k)$
$\quad \Sigma_{(k+1)h} \leftarrow U_k \Sigma_{kh} U_k$
**Return:** $\mathcal{N}(m_{Nh}, \Sigma_{Nh})$

**Algorithm 3:** Point-cloud generation

**Data:** Trained $f_\theta^{\mathrm{PC}}$, step size $h = 1/N$
**Init:** $\hat{\boldsymbol{X}}_0 = \{X_1, \ldots, X_n\} \sim \mathfrak{p}_0$
**for** $k = 0, \ldots, N-1$ **do**
$\quad \hat{\boldsymbol{X}}_{(k+1)h} \leftarrow \hat{\boldsymbol{X}}_{kh} + h f_\theta^{\mathrm{PC}}(\hat{\boldsymbol{X}}_{kh}, kh)$
**Return:** $\hat{\boldsymbol{X}}_{Nh}$

$\mu_t$ and optimal velocity field $v_t$ in the Riemannian FM objective (2), resulting in our Wasserstein FM (WFM) objective:

$$\min_\theta \int_0^1 \iint \|f_\theta^{\texttt{geo}}(\mu_t, t) - v_t\|_{L^2(\mu_t)}^2 \, \mathrm{d}\mathfrak{p}_0(\mu) \, \mathrm{d}\mathfrak{p}_1(\nu) \, \mathrm{d}t \,. \tag{10}$$

We provide a derivation of this objective function in Appendix B. As mentioned in the introduction, our two use-cases of interest are flow matching over (1) families of point-clouds, and (2) families of Gaussian distributions. While the theory outlined in Section 2.3 explicitly requires continuous distributions to ensure all objects are well-defined, the approximation of measures by point-clouds is reasonable for our applications and can be made computationally efficient courtesy of existing open-source packages (Flamary et al., 2021; Cuturi et al., 2022). In the case of Gaussian measures, the theory as described in Section 2.3.1 holds in full force. Our training algorithm is described in Algorithm 1, and Appendix E contains precise details regarding our neural network parameterization.

Finally, we mention that both frameworks can be modified by training via the *multisample FM* algorithm (Pooladian et al., 2023a; Tong et al., 2023). In brief, the idea is to augment the training regime by pairing the source and target minibatch samples according to some prescribed matching rule (instead of independent draws from both $\mathfrak{p}_0$ and $\mathfrak{p}_1$). We employ this augmentation during training, which we detail in Appendix C.

### 3.1.1 WFM OVER THE BURES–WASSERSTEIN SPACE

First suppose $\mathfrak{p}_0$ and $\mathfrak{p}_1$ are distributions over Gaussians, meaning that a batch of samples drawn from $\mathfrak{p}_0$ and $\mathfrak{p}_1$ consists of a batch of mean-covariance pairs. Here, the dynamics are straightforward: the interpolant is the McCann interpolation (recall (8)), and the velocity field over the Bures–Wasserstein manifold is also known (see (9)). Since $\mu_t$ is parameterized by $(m_t, \Sigma_t)$, the neural network is parameterized as $f_\theta^{\mathrm{BW}} : \mathbb{R}^d \times \mathbb{S}_{++}^d \to \mathbb{R}^d \times \mathbb{S}^d$, and the norm on the tangent space simplifies the computations considerably. Our final training objective becomes

$$\min_\theta \int_0^1 \iint \|f_\theta^{\mathrm{BW}}((m_t, \Sigma_t), t) - (\dot{m}_t, \dot{\Sigma}_t^{\mathrm{BW}})\|_{\mathrm{BW}(\mu_t)}^2 \, \mathrm{d}\mathfrak{p}_0(\mu) \, \mathrm{d}\mathfrak{p}_1(\nu) \, \mathrm{d}t \,. \tag{11}$$

### 3.1.2 WFM OVER DISTRIBUTIONS OF POINT-CLOUDS

In the case of point-clouds, we lose closed-form interpolations. However, we can hope to proceed so long as we have an *approximation* of the optimal transport map between the point-clouds, written $\hat{T}$. There are many works on the approximation of these maps on the basis of samples; see Hütter & Rigollet (2021); Divol et al. (2022); Manole et al. (2021); Pooladian & Niles-Weed (2021). Our goal is to have a methodology that holds for families of point-clouds of **non-uniform size**.

Figure 2: When the number of training examples is too few, all methods collapse on the training data, though our Riemannian instantiation of BW-FM captures the covariances perfectly. In the presence of sufficiently many samples, all methods generate Gaussians along the whole spiral, and our Riemannian BW-FM algorithm produces the most consistent samples. Other methods produce Gaussians with *degenerate* covariance, as the do not model geometry of the data.

We consider two approximations of optimal transport maps, both of which are based on entropic optimal transport Cuturi (2013), and are computationally efficient on GPUs due to Sinkhorn's algorithm (Sinkhorn, 1964). One approach is to round the optimal coupling to a permutation and perform the resulting interpolation. Another approach, which allows for inhomogeneous pairs of points, is to approximate $T_\star^{\mu\to\nu}$ using the entropic map (Pooladian & Niles-Weed, 2021); we provide extensive background on these objects in Appendix A, with theoretical and statistical discussions in Appendix A.3. To this end, let $X$ (resp. $Y$) represent the locations of the point-cloud $\mu \sim \mathfrak{p}_0$ (resp. $\nu \sim \mathfrak{p}_1$), and let $\hat{T}^{\mu\to\nu}$ denote the approximation of the optimal transport map. The objective (10) can be approximated by

$$\min_\theta \int_0^1 \iint \sum_i \|[f_\theta^{\mathrm{PC}}(\hat{X}_t, t)]_i - [(\hat{T}^{\mu\to\nu}(X) - X)]_i\|_2^2 \, d\mathfrak{p}_0(\mu) \, d\mathfrak{p}_1(\nu) \, dt \,, \tag{12}$$

where $\hat{X}_t = (1-t)X + t\hat{T}^{\mu\to\nu}(X)$. Here, we stress that $\hat{X}_t$ plays the role of a discretized McCann interpolation $\mu_t$. We parameterize $f_\theta^{PC}$ with a transformer and Appendix E provides further details. As both the OT map and self-attention are permutation equivariant, transformers are an organic backbone for OT based models (Haviv et al., 2024a). Moreover, unlike other point-cloud neural models such as PVCNN (Liu et al., 2019), transformers do not rely on voxelization and are not hindered by the curse-of-dimensionality.

## 3.2 GENERATION

Once $f_\theta^{\mathrm{geo}}$ is trained, we can generate new samples in a simulation-free manner as in Riemannian FM. For the Bures–Wasserstein space, we appeal to the exponential and logarithmic maps. We emphasize that the appropriate Riemannian updates are crucial to obtain non-degenerate final samples. For point-clouds, we perform a standard Euler discretization of the learned flow; see Algorithm 3.

| | BW-FM (R) | BW-FM (E) | Frobenius FM |
|---|---|---|---|
| Spiral - 16 (2D) | $\mathbf{2.98 \cdot 10^{-4}}$ | $4.00 \cdot 10^{-4}$ | $1.03 \cdot 10^{-3}$ |
| Spirals - 128 (2D) | $\mathbf{1.28 \cdot 10^{-3}}$ | $1.70 \cdot 10^{-3}$ | $2.69 \cdot 10^{-3}$ |
| Two Moons (2D) | $\mathbf{1.84 \cdot 10^{-4}}$ | $8.96 \cdot 10^{-4}$ | $1.30 \cdot 10^{-3}$ |
| Sphere (3D) | $\mathbf{6.65 \cdot 10^{-4}}$ | $2.14 \cdot 10^{-3}$ | $2.25 \cdot 10^{-3}$ |
| Cities (2D) | $\mathbf{1.88 \cdot 10^{-4}}$ | $7.26 \cdot 10^{-3}$ | $1.75 \cdot 10^{-3}$ |
| ECG (15D) | $\mathbf{9.24 \cdot 10^{-2}}$ | $3.26 \cdot 10^{-1}$ | $3.98 \cdot 10^{-1}$ |
| MERFISH (16D) | $\mathbf{1.90}$ | $1.98$ | $2.06$ |
| scRNA-seq (32D) | $\mathbf{1.31}$ | $2.74$ | $3.21$ |

Table 1: Average min. $W_2^2$ distance between each generated Gaussian and the reference datasets. Despite identical training schemes, BW-FM (R) outperforms other approaches on both synthetic and real data across several dimensions.

Car        Plane        Chair        Monitor   Vase   Toilet

Unconditional                          Conditional

Figure 3: *Left.* Synthesized samples from WFM trained on the *cars*, *planes* or *chairs* datasets. *Right.* Examples generated conditionally from the *same initial noise* via a WFM model trained on the complete 40-class ModelNet dataset.

## 4 RESULTS

### 4.1 FLOW MATCHING BETWEEN FAMILIES OF GAUSSIANS

We first demonstrate our flow matching framework between measures of Gaussian distributions on a synthetic and real datasets. For comparable baselines in each scenario, we construct two simpler flow matching approaches for Gaussian generation: (1) *Frobenius FM*, which concatenates the mean and covariances values, and trains on $\dot{\Sigma}_t^{\mathrm{E}}$ with respect to the squared-Frobenius norm, and (2) BW-FM (Euclidean), which tries to match $\dot{\Sigma}_t^{\mathrm{E}}$ but still under the BW geometry.

In all cases, we assess the quality of the learned flows by computing the the minimum distance between each generated Gaussian and the dataset using the (squared) 2-Wasserstein distance. Notably, the flows generated by Frobenius FM and BW-FM (Euclidean) do not strictly adhere to the geometry of the Bures-Wasserstein manifold, requiring synthesized covariance matrices to be artificially projected onto the space of positive semi-definite (PSD) matrices via eigenvalue truncation. In contrast, the Riemannian BW-FM algorithm consistently produces valid and accurate results across all dimensions and datasets.

### 4.1.1 TOY DATASETS

As a first test, we design a dataset of Gaussians centered on a spiral (Figure 2). When there were only few samples, only Riemannian BW-FM reconstructed the data, despite other benchmark methods following identical training regimes. On the complete 128-sample dataset, BW-FM not only reconstructs the training data, but generalizes and synthesizes novel Gaussians whose means lie on the spiral with the correct covariance profile. BW-FM shares this generalization feature with standard FM, and is able to learn the structure underlying the measure from the training data.

### 4.1.2 SINGLE-CELL GENOMICS

Spatial transcriptomics are a set of techniques which build on single-cell genomics and preserve physical information of cells' location in tissues, while assaying their gene expression. Haviv et al. (2024b) demonstrated that a cell's microenvironment can be effectively characterized using the mean and covariance of the surrounding cells gene expression. This statistical representation captures key features of cellular neighborhoods and transforms spatial transcriptomics datasets into a measure within the Bures–Wasserstein space, highlighting the value of generative modeling in this context.

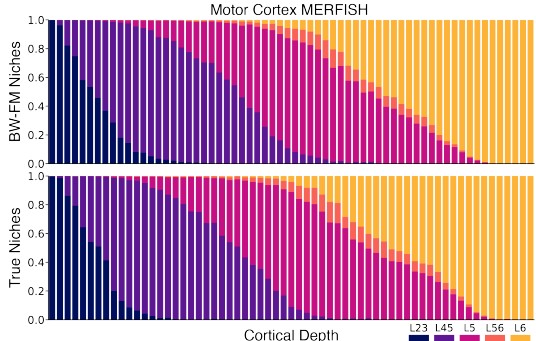

Figure 4: We apply BW-FM to realize environments of microglia from MERFISH data (Zhang et al., 2021), the composition of generated niches matches the real data along inferred cortical depth; see also Figure S2.

In the motor cortex, excitatory neurons form phenotypically distinct and highly specialized cortical layer (Zeng & Sanes, 2017). From the 254 gene MERFISH atlas (Zhang et al., 2021), we compute the mean and covariance of the top 16 principal components of gene expression from all cells within an 80 micron radius around each microglia.

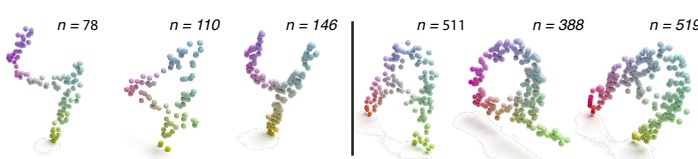

Figure 5: Generated point-clouds from MNIST and EM-NIST datasets using WFM, where $n$ denotes the number of points in each cloud.

Despite the dimensionality of this data, BW-FM synthesizes Gaussians which are highly congruent with the real data (Figure 4 and Figure S2).

Another common instance of BW manifolds arising in single-cell genomics is through aggregating cells into common states. These clusters can be summarized by their mean gene expression and its covariance. On a scRNA-seq atlas elucidating human immune response to COVID (Stephenson et al., 2021), we combine cells into MetaCells (Persad et al., 2023), and quantify the gene expression mean and covariance for each. Here too we apply BW-FM conditioned on cell-state, which encompass the heterogeneity of immune profiles appearing as response to COVID infection. Despite the plurality of labels, BW-FM can synthesize correct examples for each condition (see Figure S1).

### 4.2 FLOW MATCHING BETWEEN FAMILIES OF POINT-CLOUDS

When the two measures are point-clouds, we turn to entropic optimal transport to estimate the Mc-Cann interpolation (Cuturi, 2013). When the dataset consists of source and target point-clouds of the same size, we use a GPU-efficient rounding scheme to approximate the OT map using the entropic OT coupling; see Appendix A. When the support size of the source and target point-clouds vary, it is worth mentioning that an OT map may not even exist. Nevertheless, we approximate these curves using the entropic transport map of Pooladian & Niles-Weed (2021). Together, these approaches offer a computationally feasible solution while maintaining accuracy in transport map estimation.

We compare WFM to many other point-cloud generation algorithms. Following in their footsteps, we measure generation quality based on the 1-Nearest-Neighbour accuracy metric between generated and test-set point-clouds. On uniform, 3D datasets, WFM is competitive with current approaches (Table 3), but exemplifies itself with its unique ability to generate point-clouds with varying sizes and in high-dimensions (Table 2).

### 4.2.1 2D & 3D POINT-CLOUDS

Derived from 3D CAD designs, ShapeNet & ModelNet (Wu et al., 2015; Chang et al., 2015) are touchstone point-cloud datasets in computational geometry. Trained individually on samples from the *chair*, *car* and *plane* classes of ShapeNet, WFM synthesized high quality point-clouds with diverse profiles and matches the performance of previous 3D generation algorithms; see Figure 3 and Table 3. Our framework's versatility allows for seamless integration of label information during training, enabling the synthesis of point-clouds conditioned on specific classes. On the full 40-class ModelNet dataset, WFM learned condition dependent flows, allowing for the same initial point-cloud to generate a diverse cohort of shapes based on the desired label; see Figure 3. We stress that WFM is not restricted to only noisy source measures but can generate transformations between any two collections of point-clouds. To this end, we demonstrate that WFM can interpolate between two arbitrary elements in the dataset (e.g., between a lamp and a handbag) and complete the point-clouds based on partial profiles (e.g., generate the remaining parts of a plane); see Figure S3.

| | MNIST (4) | | Letters (A) | | seqFISH | | XENIUM | |
|---|---|---|---|---|---|---|---|---|
| | CD ↓ | EMD ↓ | CD ↓ | EMD ↓ | CD ↓ | EMD ↓ | CD ↓ | EMD ↓ |
| Current methods | NA | NA | NA | NA | NA | NA | NA | NA |
| WFM (ours) | 63.34 | 59.97 | 62.12 | 58.68 | 61.79 | 64.34 | 60.69 | 64.20 |

Table 2: 1-Nearest-Neighbour Accuracy for high-dimensional or variable size point-clouds. WFM employs a transformer backbone and relies on the efficient computation of the entropic transport map, allowing it to scale to arbitrary dimensions and learn flows between point-cloud of variable sizes, key features all previous point-cloud generation approaches lack.

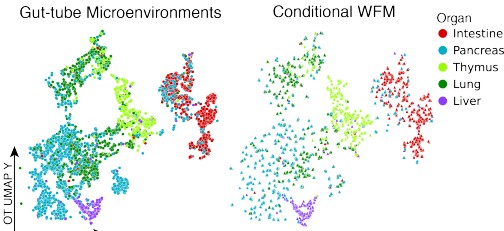

Figure 6: The niche of each cell is the point-cloud from gene-expression profiles for cells in its environments. Using the top 16 principal components of gene expression, WFM generates high-dimensional microenvironments of gut-tube cells based on the gastrulating organs.

Another novel facet of WFM is its ability to perform generative modeling from inhomogeneous datasets, where the number of points varies between independent samples. This happens in the MNIST or Letters datasets, where data is generated by thresholding grayscale numerical values. In this setting, we use the entropic transport map to approximate the objective; see (14). WFM sets itself apart from other methods, which are restricted to uniform datasets, by leveraging the entropic OT map's ability to compute feasible transformations between point-clouds of different sizes. Our experiments in Figure 5 demonstrate that WFM generates high-quality & diverse samples, despite large variability in the number of points per sample, which is itself a novel contribution.

### 4.2.2 Spatial Transcriptomics (High Dimensional Point Clouds)

In spatial transcriptomics, the niche of a cell is the point-cloud in high-dimensional gene-expression space of its immediate nearest neighbours. This approach is complementary to the BW representation of a niche (recall Section 4.1.2), and serves as a more high fidelity view suited for fine-grain interactions. Due to their high dimensionality, cellular microenvironments have remained beyond the reach of point-cloud-based generative models that depend on voxel-based neural networks. Instead, WFM uses transformers, which due to their permutation equivariance and indifference to dimensionality, are natural architectures for spatial transcriptomics point-clouds (Haviv et al., 2024b).

During embryogenesis, specific regions within the primitive gut tube differentiate into organs such as the liver or lungs based on interactions between the gut and surrounding mesenchyme (Nowotschin et al., 2019). Applied on environments of gut-tube cells from a seqFISH dataset of mouse embryogenesis (Lohoff et al., 2022), WFM synthesized cellular niches conditioned on organ labels, thus de-

|  | Airplane | | Chair | | Car | |
|---|---|---|---|---|---|---|
|  | CD ↓ | EMD ↓ | CD ↓ | EMD ↓ | CD ↓ | EMD ↓ |
| PointFlow | 75.68 | 70.74 | 62.84 | 60.57 | 58.10 | 56.25 |
| SoftFlow | 76.05 | 65.80 | 59.21 | 60.05 | 64.77 | 60.09 |
| DPF-Net | 75.18 | 65.55 | 62.00 | 58.53 | 62.35 | 54.48 |
| Shape-GF | 80.00 | 76.17 | 68.96 | 65.48 | 63.20 | 56.53 |
| PVD | 73.82 | 64.81 | 56.26 | 53.32 | 54.55 | 53.83 |
| PSF | 71.11 | 61.09 | 58.92 | 54.45 | 57.19 | 56.07 |
| WFM (ours) | 73.45 | 71.72 | 58.98 | 57.77 | 56.53 | 57.95 |

Table 3: Using 1-Nearest-Neighbour Accuracy based on Earth Mover's Distance (EMD) and Chamfer's Distance (CD). Wasserstein Flow Matching (WFM) is competitive with existing approaches (data from Wu et al. (2023)), while producing diverse samples, see Figure 3.

manding an understanding of the interplay between spatial context and phenotype. Despite the intricate nature of the gastrulation process, compunded by the dataset's dimensionality, WFM can accurately generate organ-specific niches; see Figure 6 and Table 2.

## 5 Conclusion and outlook

This work shows how to appropriately *lift* the Riemannian flow matching paradigm of Chen & Lipman (2023) to the Wasserstein space, resulting in Wasserstein flow matching. Our motivations stem from modern datasets, where each sample of data can itself be viewed as a probability distribution, necessitating this extension for generative modeling purposes. Our contributions are algorithmic in nature, which incorporate various elements, such as estimating optimal transport maps via entropic optimal transport, closed-form expressions over the Bures–Wasserstein space, and attention mechanisms in neural network architectures. Our algorithm is capable of generating realistic data from Gaussian and variable-size or high-dimensional point-clouds. Both contexts are highly relevant in single-cell and spatial transcriptomics for synthesizing of microenvironments and cellular states.

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

# A    ENTROPIC ESTIMATION OF OT MAPS

We briefly discuss how to estimate optimal transport maps between point-clouds using entropic optimal transport. We refer the interested reader to Pooladian & Niles-Weed (2021) for more information on this approach.

We first outline the numerical aspects of the approach; we follow Peyré & Cuturi (2019). Let $\mu = \sum_i m^{-1}\delta_{x_i}$ and $\nu = \sum_j n^{-1}\delta_{y_j}$, where $\boldsymbol{X} = \{x_1, \ldots, x_m\}$, $\boldsymbol{Y} = \{y_1, \ldots, y_n\}$. We first define the following polyhedral constraint set

$$U_{m,n} \coloneqq \left\{ P \in \mathbb{R}_+^{m \times n} \; : \; P\mathbf{1}_m = m^{-1}\mathbf{1}_m \,, P^\top \mathbf{1}_n = n^{-1}\mathbf{1}_n \right\},$$

which represents the possible couplings between the two discrete measures. The entropic optimal transport coupling between the two discrete measures $\mu$ and $\nu$ is defined as the minimizer to the following strictly convex optimization problem

$$\boldsymbol{P}^\star \coloneqq \operatorname*{argmin}_{P \in U_{m,n}} \langle C, P \rangle + \varepsilon H(P) \,, \tag{13}$$

where $\varepsilon > 0$, $H(P) \coloneqq \sum_{i,j} P_{i,j}(\log(P_{i,j}) - 1)$, and $C_{i,j} \coloneqq \|x_i - y_j\|_2^2$. Sinkhorn's matrix scaling algorithm (Sinkhorn, 1964) makes it possible to solve for $\boldsymbol{P}^\star$ with a runtime of $O(mn/\varepsilon)$ (Altschuler et al., 2017). We briefly stress three points:

1. The coupling $\boldsymbol{P}^\star$ is *not* a permutation matrix. The coupling lies inside the polytope $U_{m,n}$ and not at the vertices, and therefore is not a permutation matrix.

2. When $\varepsilon = 0$, the objective becomes a standard linear program with a runtime of $\tilde{O}(mn(m + n))$ (up to log factors) (Peyré & Cuturi, 2019, Chapter 3). While we include a CPU implementation (Flamary et al., 2021) in the WFM codebase, this approach lacks GPU efficiency and substantially increases training time, making it impractical for most use cases.

3. Instead, the regularization parameter $\varepsilon$ serves as a tunable training hyperparameter. Lower $\varepsilon$ values better approximate true the optimal transport map but require more Sinkhorn iterations for convergence, creating a direct trade-off between accuracy and computational efficiency.

In all our experiments, we used the open-source package OTT-JAX[3] to compute the entropic coupling and the out-of-sample mapping (Cuturi et al., 2022).

## A.1    ROUNDED MATCHINGS

Our first approach holds when $m = n$. In this case, we can greedily *round* the noisy matching matrix $\boldsymbol{P}^\star$ to become a permutation. This is achieved through an iterative process of selecting the maximum value (argmax) and zeroing out corresponding rows and columns. This method repeatedly identifies the largest remaining probability, sets it to 1, and eliminates other entries in its row and column, ultimately resulting in a permutation matrix that preserves the probabilistic assignment implied by the original doubly stochastic matrix. This is merely a GPU-friendly heuristic approximation to the true optimal permutation matrix between the two point-clouds.

## A.2    ENTROPIC TRANSPORT MAP: AN OUT-OF-SAMPLE ESTIMATOR

A primal-dual relationship of the strictly convex program (13) shows that there exist vectors $(\boldsymbol{f}^\star, \boldsymbol{g}^\star) \in \mathbb{R}^m \times \mathbb{R}^n$ such that

$$\boldsymbol{P}_{i,j}^\star = e^{\boldsymbol{f}_i^\star/\varepsilon} e^{-\boldsymbol{C}_{i,j}/\varepsilon} e^{\boldsymbol{g}_j^\star/\varepsilon}$$

These two vectors are called the Kantorovich potentials, which are initially defined on the support of $\mu$ and $\nu$, respectively. However, they can be readily extended to all of $\mathbb{R}^d$ (Mena & Niles-Weed,

---

[3]See https://ott-jax.readthedocs.io/en/latest/.

2019), resulting in two functions

$$\hat{f}(x) = -\varepsilon \log\Big(\sum_{j=1}^{n} n^{-1} \exp((\boldsymbol{g}_j^\star - \|x - y_j\|^2)/\varepsilon)\Big),$$

$$\hat{g}(y) = -\varepsilon \log\Big(\sum_{i=1}^{m} m^{-1} \exp((\boldsymbol{f}_i^\star - \|y - x_i\|^2)/\varepsilon)\Big).$$

Following Pooladian & Niles-Weed (2021), we can define the *entropic transport map*, where the last equality is a simple calculation:

$$\hat{T}_\varepsilon(x) := x - \nabla \hat{f}(x) = \frac{\sum_{j=1}^{n} y_j \exp((\boldsymbol{g}_j^\star - \|x - y_j\|^2)/\varepsilon)}{\sum_{j=1}^{n} \exp((\boldsymbol{g}_j^\star - \|x - y_j\|^2)/\varepsilon)}. \tag{14}$$

This estimator was initially to provide statistical approximations to the optimal transport map $T_\star^{\mu \to \nu}$ on the basis of samples; see Pooladian & Niles-Weed (2021); Pooladian et al. (2023b; 2022). Note that $\hat{T}_\varepsilon(x)$ can be interpreted as the conditional expectation of the plan $\boldsymbol{P}^\star$ conditioned on out-of-sample inputs $x \in \mathbb{R}^d$, which is well-defined due to the relations above. Finally, we stress that this estimator can be adapted to settings where the point-clouds $\mu$ and $\nu$ not only have different numbers of points, but also non-uniform weights. As this estimator is also a by-product of Sinkhorn's algorithm, it is also scalable and GPU-friendly.

### A.3 On the (statistical) approximations of geodesics

We briefly collect a basic results pertaining to the (statistical) approximation of optimal transport paths. This bound shows that the error grows along the trajectory, but is limited by the overall distance of the maps.

**Proposition A.1.** *Let $\mu, \nu$ be two probability measures and suppose $\mu$ has a density, and let $T^\star$ be the optimal transport map from $\mu$ to $\nu$. Let $\hat{T}$ be an estimator to the optimal transport map, defined with respect to data $X_1, \ldots, X_n \sim \mu$ and $Y_1, \ldots, Y_n \sim \nu$. Then for $t \in [0, 1]$*

$$\mathbb{E}[W_2^2(\rho_t, \hat{\rho}_t)] \leq t^2 \mathbb{E}\|\hat{T} - T^\star\|_{L^2(\mu)}^2,$$

*where the outer expectation is taken with respect to the data, and we define*

$$\rho_t := ((1-t)\mathrm{id} + tT^\star)_\sharp \mu, \quad \hat{\rho}_t := ((1-t)\mathrm{id} + t\hat{T})_\sharp \mu$$

*Proof.* The result follows immediately from a standard coupling argument to obtain the linearized Wasserstein distance (Wang et al., 2010; Panaretos & Zemel, 2020)

$$W_2^2(\rho_t, \hat{\rho}_t) \leq \|((1-t)\mathrm{id} + tT^\star) - ((1-t)\mathrm{id} + t\hat{T})\|_{L^2(\mu)}^2 = t^2 \|\hat{T} - T^\star\|_{L^2(\mu)}^2. \qquad \square$$

The entropic Brenier map is one particular estimator. We note two key properties of this map; see Pooladian & Niles-Weed (2021) for in-depth discussions.

**Theorem A.2.** *Suppose $\mu, \nu$ have density bounded above and below, and that the optimal transport map between them, denoted $T^\star$, is such that $(T^\star)^{-1}$ is at least twice differentiable and there exists $\lambda, \Lambda > 0$ such that*

$$\lambda I \preceq DT^\star \preceq \Lambda I.$$

*Then, when estimated from $n$ samples from $\mu$ and $n$ samples from $\mu$, the entropic Brenier map has the following error*

$$\mathbb{E}\|\hat{T}_\varepsilon - T^\star\|_{L^2(\mu)}^2 \lesssim n^{-1/2} \log(n) \varepsilon^{-d/2-1} + \varepsilon^2, \tag{15}$$

*where we suppress constants that depend on our assumptions. Performing a bias-variance trade-off in the regularization parameter, one obtains $\varepsilon = \varepsilon(n) \asymp n^{-1/(d+4)}$ and the total error becomes*

$$\mathbb{E}\|\hat{T}_\varepsilon - T^\star\|_{L^2(\mu)}^2 \lesssim_{\log(n)} n^{-2/(d+4)}.$$

We emphasize that the assumptions in Theorem A.2 are standard in the literature (Hütter & Rigollet, 2021; Deb et al., 2021; Muzellec et al., 2021; Divol et al., 2022). While the rate scales exponentially poorly with the dimension, we stress that existing lower bounds of estimation (see Hütter & Rigollet (2021)) also suffer from the curse of dimensionality, which is unavoidable for this task. Combining these two results, we can compare the geodesic given by the OT map, and the one induced by using the entropic map, where we write

$$\hat{\rho}_t^\varepsilon := ((1-t)\mathrm{id} + t\hat{\mathrm{T}}_\varepsilon)_\sharp \mu \,.$$

**Corollary A.3.** *Consider the same setting as Theorem A.2. Then the geodesic given by the estimated entropic Brenier map, denoted by $\hat{\rho}_t^\varepsilon$, on the basis of $n$ samples and $\varepsilon \asymp n^{-1/(d+4)}$, is close to the true geodesic with the following error*

$$\mathbb{E}[W_2^2(\rho_t, \hat{\rho}_t^\varepsilon)] \lesssim t^2 n^{-2/(d+4)} \,.$$

## B  DERIVATION OF THE WFM OBJECTIVE

In this section, we give validity to the WFM for optimization purposes, and our choice of curves. For instance, recall the original Flow Matching objective (Lipman et al., 2022)

$$\mathcal{L}_{\mathrm{FM}}(\theta) := \int_0^1 \mathbb{E}_{X_t \sim p_t} \|f_\theta(X_t, t) - u_t(X_t)\|^2 \, dt \,, \tag{16}$$

where $f_\theta : \mathbb{R}^d \times [0,1] \to \mathbb{R}^d$ is a neural network, and $(p_t, u_t)_{t \in [0,1]}$ are a density-vector field pair that satisfy the continuity equation between two distribution $\mu$ and $\nu$.

$$\partial p_t + \nabla \cdot (p_t u_t) = 0 \,, \quad p_0 = \mu, p_1 = \nu \,.$$

Note that, implicit in $\mathcal{L}_{\mathrm{FM}}(\theta)$ is the endpoint constraints $\mu$ and $\nu$. Now, we average over possibly choices of $\mu \sim \mathfrak{p}_0$ and $\nu \sim \mathfrak{p}_1$, resulting in

$$\int_0^1 \iint \mathbb{E}_{X_t \sim p_t} \|f_\theta(X_t, t) - u_t(X_t)\|^2 \, d\mathfrak{p}_0(\mu) \, d\mathfrak{p}_1(\nu) \, dt \,. \tag{17}$$

As a particular case, take $(p_t, u_t) \leftarrow (\mu_t, v_t)$, where the first argument is the McCann interpolation between $\mu$ and $\nu$, and $v_t$ is the optimal velocity field, which is a function of the optimal transport map from $\mu$ to $\nu$ (recall Section 2.3). This yields our final objective (10), which we recall here for convenience

$$\mathcal{L}_{\mathrm{WFM}}(\theta) := \int_0^1 \iint \mathbb{E}_{X_t \sim \mu_t} \|f_\theta(X_t, t) - v_t(X_t)\|^2 \, d\mathfrak{p}_0(\mu) \, d\mathfrak{p}_1(\nu) \, dt \,. \tag{18}$$

When $\mathfrak{p}_0, \mathfrak{p}_1$ are distributions over Gaussians, we have closed-form expressions for all objects of interest. When $\mathfrak{p}_0, \mathfrak{p}_1$ are distributions over point-clouds, we approximate the geodesics between the points using entropic Brenier maps and their respective interpolations. We emphasize that the rounded-matching which we employ (Appendix A.1) is also a valid curve.

## C  MULTISAMPLE WASSERSTEIN FLOW MATCHING

Since optimal transport can be applied on the Wasserstein manifold itself, both WFM and BW-FM can be seamlessly integrated with the multisample FM (MS-FM) framework (Pooladian et al., 2023a; Tong et al., 2023). The core technique behind MS-FM is to use OT to match minibatches from source and target measures during training, rather than relying on random pairings. This has shown to improve learned flows while requiring fewer function evaluations to synthesize new samples. Applying MS-FM requires computing the pairwise distance matrix between source and target batch samples, denoted from $i \in \{1, \ldots, \mathtt{Bsz}\}$. In the BW-FM setting, given two sets of Gaussians $\{(a_i, A_i)\}_{i=1}^{\mathtt{Bsz}}$ and $\{(b_i, B_i)\}_{i=1}^{\mathtt{Bsz}}$, their Frechét ($W_2^2$) distance matrix is:

$$C_{i,j} = \|a_i - b_j\|_2^2 + \mathrm{Tr}(A_i + B_j - 2(A_i^{1/2} B_j A_i^{1/2})^{1/2}) \tag{19}$$

We then use entropic OT to approximately solve the assignment problem on $C$ and compute a transport matrix. This is the converted into a one-to-one assignment matrix via rounded matching (Appendix A.1), ensuring the entire batch is used in training.

For WFM point-clouds, applying MS requires computing pairwise OT distance between all source and target samples within a minibatch. For large point-clouds, this is exorbitantly expensive, even with Sinkhorn iterations. For an efficient approximate, here too we rely on the Frechét distance, computed between empirical means and covariances of each point-cloud. Computation of the Frechét distance is markedly less resource-intensive than entropic OT, yet is notably correlated with EMD values (see Table 1 in (Haviv et al., 2024b)).

## D  SAMPLING FROM SOURCE MEASURE

In both WFM and BW-FM, learning flows requires a source measure which is straightforward to sample from. For a source distribution on the space of $\{(m, \Sigma) : m \in \mathbb{R}^d, \Sigma \in \mathbb{S}_{++}^d\}$, we simply sample means and covariance matrices using independent Gaussian and Wishart distributions, respectively. By default, the parameters for the Gaussian component of the source matches the average and standard deviation of the means in the target, while the scale parameter in the Wishart is the barycenter of the data covariance.

To achieve high-quality generation of point-clouds, it is essential that the initial (source) distribution be diverse, rather than collapsed and degenerate. Indeed, while it is alluring to produce noisy point-clouds by sampling points from a single base distribution, i.e. $X = \{x_i\}_{i=1}^n, x_i \sim \mathcal{N}(0, I_d)$, as $n$ grows, the Wasserstein distance between instances goes to $0$. To alleviate this, we draw point-clouds from multivariate Gaussians with a stochastic covariance:

$$L \sim \mathcal{N}(\mu_L, \sigma_L \cdot I)$$
$$X = \{x_i\}_{i=1}^n, x_i \sim \mathcal{N}(0, LL^T)$$

where $\mu_L$ & $\sigma_L$ are the average and standard deviation of the Cholesky factors from the empirical covariances of the target measure point-clouds. This ensures a wider source measure, producing a diverse range of noise point-clouds.

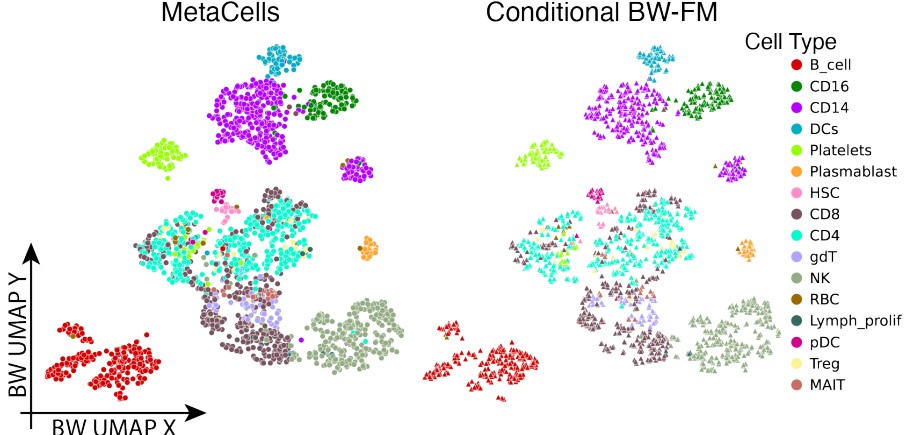

Figure S1: Conditional BW-FM applied to single-cell RNA sequencing data of immune response to COVID-19. Large scale single-cell atlases are commonly grouped into highly dedicated clusters called MetaCells (Persad et al., 2023). In this application, BW-FM is conditioned on cell state and trained to generate means and covariances of gene expression, focusing on the top 32 principal components, derived from aggregated cells. The model achieves high-quality sample generation, as evidenced by a label accuracy of 93.13%.

# E  NEURAL ARCHITECTURE & TRAINING

## E.1  BW-FM ON GAUSSIANS

The goal of BW-FM is to train a neural network to match the (Riemannian) time-derivative along the BW geodesics between Gaussians. The model employs a standard, fully connected neural network which takes as input concatenated values of $(m_t, \Sigma_t, t)$ based on the McCann interpolation formula from Section 2.3.1. Since the covariance matrix is symmetric, only its lower-diagonal values are used, flattened into a vector of length $d(d+1)/2$. Time values are converted to Fourier features, an approach inspired by positional encodings in transformer literature (Vaswani, 2017). To streamline training, two separate networks are employed: one to match the time derivative of the mean $\dot{m}_t$ and another for the time derivative of the covariance matrix $\dot{\Sigma}_t^{\mathrm{BW}}$. The BW tangent norm is used as the loss function for training these networks.

By default, all models use a 6-layer neural network using *relu* non-linearity, with 1024 neurons per layer, applying skip connections and layer-norm Ba (2016). Training is performed for 100,000 gradient descent steps using the Adam optimizer (Kingma, 2014) with an exponential learning rate decay of 0.97 every 1000 steps and batch size of 128.

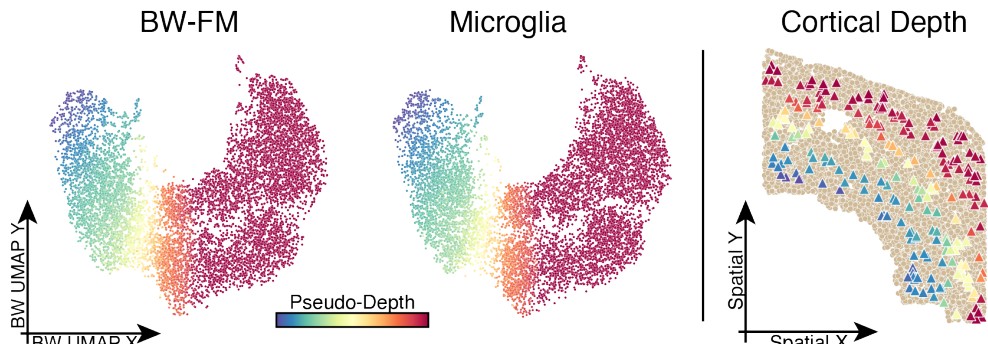

Figure S2: Spatial arrangement of microglia in the motor cortex. Bures-Wasserstein distance based 2D UMAP visualization of real microglia and BW-FM synthesized niches, colored by their first diffusion component (DC). This DC corresponds to the cortical depth of the microglia across the MERFISH slices.

## E.2  WFM ON POINT-CLOUDS

WFM is designed to estimate the optimal transport (OT) map for a given pair of interpolate point-cloud and time $(\boldsymbol{X}_t, t)$. Here too the time component $t$ is first converted into Fourier features. The model's architecture begins with an embedding layer, followed by a series of alternating multi-head attention and fully-connected layers. Skip connections and layer-norm are applied after each operation. The final layer projects the embeddings back to $X$'s original space using a dense layer with zero initialization. The model is trained by minimizing the squared distance between the predicted and true OT maps.

By default, the entropic OT map is constructed with regularization weight of $\varepsilon = 0.002$ and 200 Sinkhorn iterations, which we found to be sufficient for convergence. Whenever the dataset consists of uniformly sized point-clouds, we use rounded matching (Appendix A.1), otherwise we apply the out-of-sample estimator (Appendix A.2) which can calculate maps between point-clouds with different sizes. The transformer network is composed to 6 multi-head attention blocks, with an embedding dimension of 512 and 4 heads. Our model is optimizer with Adam (Kingma, 2014) using an exponential learning rate decay and batch size of 64.

WFM relies on JAX and OTT-JAX (Bradbury et al., 2021; Cuturi et al., 2022) and enjoys seamless optimization via end-to-end just-in-time compilation. For the ShapeNet experiments (Table 3), the model is trained for 500,000 training steps, totaling to about 3 days of trainings of a single A100 GPU. All other experiments (Table 2) trained for 100,000 steps, requiring around 3-4 hours of GPU use. We note the Transformer's forward and backwards pass was the most significant source

of computational overhead, as opposed to the Sinkhorn based approximation of OT maps. The computational complexity of both components is quadratic with point-cloud size, which limits the scalability of WFM. Making Transformers simpler to optimize and accelerating OT computations are both active areas within ML research (Amos et al., 2022; Scetbon et al., 2021; Zhang et al., 2022), and future iterations of WFM can incorporate solution from those spaces.

## F  EXPERIMENT DETAILS

### F.1  SPATIAL TRANSCRIPTOMICS

In our manuscript, we applied WFM and BW-FM on several spatial transcriptomics datasets, encompassing a variety of technologies and tissue contexts. From a 254-gene MERFISH atlas of the motor cortex (Zhang et al., 2021), we focus on niches of microglia cells. We compress gene-expression profiles down to their 16 principal components (PC) and aggregate all the cells around each microglia within an 80 micron radius, yielding on average 26.6 cells per niche. We then calculated the gene-expression PC mean and covariance within each environment to produce Gaussians for BW-FM. Generated Gaussians align with real data and span for the full cortical depth of the microglia niches (Figure S2). In Figure 4, we predict the environment composition by cell type for generated Gaussians via nearest-neighbour regression in BW space using real data as supervision, demonstrating congruence between the two across cortical depth.

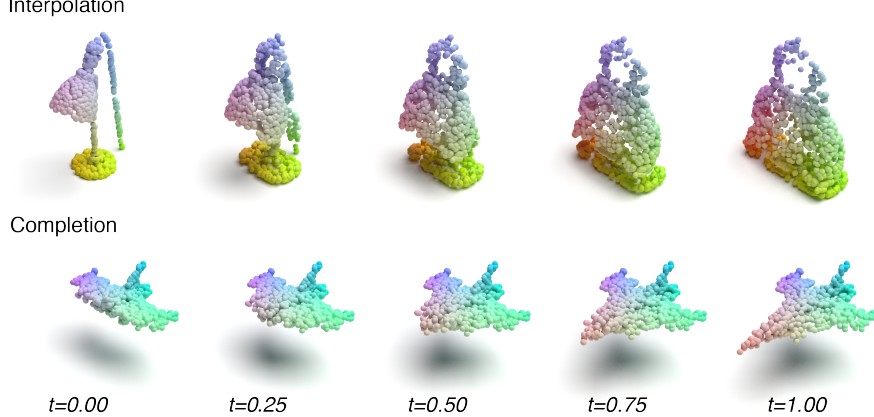

Interpolation

Completion

t=0.00          t=0.25          t=0.50          t=0.75          t=1.00

Figure S3: Interpolation and shape completion with WFM. *Top*. Using the *lamps* and *handbags* as the source and target measures, WFM learns to transform a given (unseen test-set) lamp point-cloud into a valid handbag. *Bottom*. Trained to generate full planes, WFM can reconstruct complete point-clouds from partial views of test-set samples.

In a complementary approach, WFM is applied directly on gene-expression based point-clouds of niches, and does not require the Gaussian representation. Uniquely suited for high-dimensional data, we apply WFM on seqFISH assay of embryogenesis (Lohoff et al., 2022) and a XENIUM experiment of melanoma metastasis to the brain (Haviv et al., 2024b). In both dataset, we select the $k = 8$ physical nearest neighbours of every cell, and aggregate their first 16 PCs to produce environment point-clouds.

From the seqFISH dataset, we concentrated on the gut-tube region, which is divided into spatially segregated, gastrulating organs. Applied unconditionally, WFM generated niches match the distribution of the real data based on EMD and CD 1-nearest-neighbour accuracy (Table 2). We then assessed WFM's capability to comprehend the relationship between cell state and environment and tasked it with conditional generation based on organ label. Based on OT distances estimated via *Wormhole* embeddings (Haviv et al., 2024a), point-clouds from WFM recapitulated true organ environment.The label accuracy for WFM-generated data was 78.86%, which was nearly identical to the test-set real data accuracy of 79.59%.

# G  2D & 3D POINT-CLOUDS

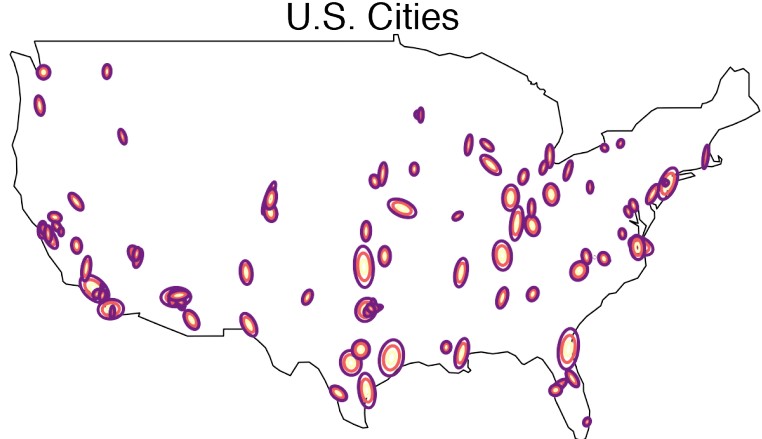

Figure S4: Cities Dataset. Gaussians representing the $100$ most populous cities in the continental US. The data was obtained from Bennett (2010) via OSMnx (Boeing, 2017). The mean parameter is the longitude and latitude coordinate of each city and the covariance is the 2nd moment approximation of their metro area.

The ShapeNet dataset consists of 3D point-clouds of $55$ different classes, each one comprised of $15,000$ points. Emulating the benchmarking effort in (Wu et al., 2023), we apply WFM to generate $n = 1000$ sized examples from the *plane*, *car* and *chair* classes. At each gradient descent step, we sample $64$ point-clouds from the training set for each class, and randomly select $n = 1000$ points from each. To evaluate generation quality, we synthesize point-clouds to much the size of the test set, and calculate the real or generated $1 - NN$ accuracy based on EMD and CD metrics.

ModelNet has $40$ classes of point-clouds, with $2048$ points in each. Conditioned on class label, WFM is trained to generate $n = 1000$ sized point-clouds here too. In this setting, the noise measure is the standard normal and we did not use multi-sample matching. According to nearest-neighbour classification from OT preserving *Wormhole* embeddings, generated samples match their class with an accuracy of $77.66\%$, approaching the $79.98\%$ purity of test set samples from real-data.

The MNIST dataset is a widely used collection of handwritten digits, consisting of 28x28 pixel grayscale images of the numbers 0 through 9. EMNIST (Extended MNIST) is an expansion of MNIST that includes handwritten letters as well as digits. To convert samples from these datasets into point-clouds, we threshold each image and extract the coordinates of the above-threshold pixels. This produces a cohort of point-clouds of variables sizes, as each image contains a different number of relevant pixel. We apply the entropic OT map (see Appendix A) based WFM to synthesize point-clouds of the digit $4$ and letter $a$. Despite the data heterogeneity, WFM produces realistic examples (Figure 5), while capturing the data distribution (Table 2). We again stress that this is a unique feature of WFM, lacking from any previous point-cloud generation algorithm

