# OpenReview forum: "Wasserstein Flow Matching: Generative modeling over families of distributions"
_ICLR.cc/2025/Conference — Submitted to ICLR 2025_

### Official Review · Reviewer_UUcS · 2024-11-02

**Soundness:** 3
**Presentation:** 3
**Contribution:** 2
**Rating:** 6
**Confidence:** 4

**Summary:**

This paper introduces Wasserstein Flow Matching (WFM) for families of set of particles and Gaussians. In other words, this paper attempts to train the FM model on Wasserstein space. For given the pairs of probability densities $\Gamma$, the method first approximates McCann’s interpolant of $(\mu, \nu) \sim \Gamma$ through traditional OT algorithms or closed-form solutions. By this approximation, they obtain the interpolant $\mu_t$, and the velocity $v_t$. Then, the approach trains the model $F_\theta$ that learns the mean field on each $(t, \mu_t)$ through flow matching loss ($\\| F_\theta (t, \mu_t) - v_t \\|^2$). This paper evaluates their algorithm on the cell dataset and point cloud generation.

**Strengths:**

- The paper presents an new concept of Wasserstein Flow Matching, which extends the Flow Matching framework to probability distribution spaces.
- It effectively incorporates Gaussian cases through McCann’s (displacement) interpolant.
- The method is applied to a variety of data types, including genomic data and point cloud data.
- Overall, the paper is clear, well-structured, and easy to follow.

**Weaknesses:**

- The motivation for learning the mean field on the Wasserstein space is not entirely clear, as the mean field may not capture meaningful structures like Wasserstein geodesics. Additionally, the approach first estimates McCann’s interpolation using traditional methods, followed by the FM algorithm. This sequential combination of the two algorithms feels somewhat incremental in terms of contribution.

- For cases where families of particle sets (point cloud) are considered, the paper estimates entropic OT, which deviates from the theoretical framework. Calculating the McCann's interpolant $\mu_t$ requires estimating the standard OT rather than entropic OT.

- Even with reduced computational overhead through estimating entropic OT, I believe there would be still huge computational burden when estimating $\mu_t$ in point cloud application. It might be helpful if the authors could specify the computational costs associated with solving entropic OT during training. I believe there would be scalability issues when the number of particles becomes more than 2000.

- Overall, the method involves an inaccurate approximation (e.g. standard OT -> entropic OT) and seems to be computationally demanding (e.g. sinkhorn for every iterations). Moreover, its performance results is not convincing for me, too. I believe the benefit of WFM method and the potential applications should be more carefully carried.

**Questions:**

- Could the authors provide training times, including times for solving entropic OT, for the point cloud generation tasks? Additionally, a comparison with other models, such as diffusion-based point cloud generation methods, would be insightful.

- Does the optimal velocity field $v_t$ that minimizes WFM objective guarantee the transportation of $\mathfrak{p}_0$ to $\mathfrak{p}_1$? Could the authors provide the theoretical proofs?

- What is the difference of this algorithm compare to mini-batch OT with Wasserstein distance cost (in point clouds)?

$ $

**Additional Comments** Thank you for your response. However, certain aspects of the authors' explanation remain unclear. First, the authors assume that the marginals must have a "pdf." Furthermore, they claim to apply this assumption to Flow Matching, which requires a positive density across the entire domain (generally, I think the domain is $\mathcal{W}_{2,ac}(M)$). I believe these two assumptions are contradictory since it is (typically) not possible to define a pdf in infinite-dimensional spaces; only distributions exist in such cases. This seems to create a fundamental inconsistency in the assumptions.

That said, I do understand and appreciate some points the authors have discussed. For instance, the Bures-Wasserstein manifold is a finite-dimensional Riemannian manifold, which allows for the direct application of Riemannian Flow Matching. Consequently, I understood that we can define a "nice subspace" that ensures the flow matching vector field $v$ can transport one marginal to another. With this clarification, the discussion becomes much clearer.

I believe that remaining small issues can be easily addressed, so I will raise my score to 6.

---

> ### Author Response · Authors · 2024-11-19
> **Response to Reviewer UUcS:**
>
> Thank you for your insightful remarks and thoughtful questions.
>
> > The paper presents a new concept of Wasserstein Flow Matching, which extends the Flow Matching framework to probability distribution spaces. It effectively incorporates Gaussian cases through McCann’s (displacement) interpolant. The method is applied to a variety of data types, including genomic data and point cloud data. Overall, the paper is clear, well-structured, and easy to follow.
>
> We are grateful for the reviewer’s positive remarks of our work, specifically regarding the conceptual novelty of WFM and the manuscript's clarity.
>
> > The motivation for learning the mean field on the Wasserstein space is not entirely clear, as the mean field may not capture meaningful structures like Wasserstein geodesics.
>
> There is no mention of “mean field” in our paper — would the reviewer be able to clarify this point so we can appropriately address it?
>
> The reviewer is correct that flowing along the Wasserstein geodesic is a critical component of our algorithm. Indeed, WFM does exactly that as our method explicitly learns
>
> 1.  For Gaussians: The vector fields in equations 8-9 exactly represent the Wasserstein geodesic paths.
>
> 2.  For point-clouds: The full OT solution yields the true Wasserstein geodesic, which we efficiently approximate using entropic regularization with minimal loss in accuracy.
>
> These theoretical aspects are now discussed in depth in our revision (see **Appendix A.3** Appendix **B**).
>
> > Additionally, the approach first estimates McCann’s interpolation using traditional methods, followed by the FM algorithm. This sequential combination of the two algorithms feels somewhat incremental in terms of contribution.
>
> Along with other impactful generative models for statistical domains, such as [Fisher](https://arxiv.org/abs/2405.14664), [Dirichlet](https://arxiv.org/abs/2402.05841), [Categorical](https://arxiv.org/abs/2405.16441) and [SE(3) Flow Matching](https://arxiv.org/abs/2310.02391) [1,2,3,4],  our work is also a variant of Riemannian Flow Matching (RFM). Our contribution is specifically lifting RFM to the Wasserstein space, which seamlessly models both families of Gaussian distributions and families of point-clouds.
>
> Working in the space of distributions required several crucial innovations, such as modeling point-clouds with a transformer to capture the permutation equivariance of the OT map. We believe this is a surprising novelty that has yet to be explored in many works at the intersection of machine learning and optimal transport, with future applications abundant. The transformer combined with the entropic OT estimate allows for generation of variably-sized and high-dimensional point-clouds, which to the best of our knowledge makes WFM the first example of a model capable of this.
>
> > For cases where families of particle sets (point cloud) are considered, the paper estimates entropic OT, which deviates from the theoretical framework. Calculating the McCann's interpolant μt requires estimating the standard OT rather than entropic OT.
>
> We acknowledge this deviation from the theoretical framework. Our use of entropic OT is a practical choice that enables GPU training, unlike unregularized OT. When combined with rounded matching (**A.1** in the appendix), the entropic estimate still produces valid vector fields and adheres to flow-matching theory. We discuss additional details of this in **Appendix  A** and **Appendix B**, and our general response above. These sections in the appendix discuss how the regularization parameter offers a trade-off between theoretical exactness and computational efficiency, as smaller values require more Sinkhorn iterations.
>
> For completeness, we've added support for full, unregularized OT via [POT](https://pythonot.github.io/), but it is significantly slower and **cannot handle point-clouds of different sizes**.

---

> ### Author Response · Authors · 2024-11-19
> **Response to Reviewer UUcS:**
>
> > Even with reduced computational overhead through estimating entropic OT, I believe there would be still huge computational burden when estimating μt in point cloud application. It might be helpful if the authors could specify the computational costs associated with solving entropic OT during training.
>
> We agree with the reviewer that there is a non-trivial computational overhead required to estimate the OT map, even in the entropic setting. As noted, WFM provides users with the ability to tune between theoretical adherence and computational efficiency via the entropic regularization parameter.
>
> There are two sources of computational burden in our model. The first, as the reviewer noted, is the computational estimation of the OT map, and the second is the transformer and its backward pass. Surprisingly, we found that it is the transformer which consumes more resources, not the OT estimation. Both accelerating the computation of OT maps and increasing the efficiency of transformers are highly active areas of research within ML. We envision future work incorporating some of these solutions, namely [MetaOT](https://arxiv.org/abs/2206.05262) and [SetTransformers++](https://proceedings.mlr.press/v162/zhang22ac.html) [5,6] for streamlined training of WFM.
>
> For the ShapeNet results in **Table 3** and **Figure 3**, training consisted of 500,000 gradient descent steps with a batch size of 64 which required about 3 days on a single A100 GPU. WFM on datasets with fewer points per sample, such as those in Figures 5-6 and Table 2, trains much faster, requiring only 3-4 hours. We note these training parameters were not optimized, and it is entirely possible that comparable performance can be reached in fewer iterations or a smaller batch size. We have added an extended discussion on computational requirements in the revised manuscript (**Appendix E**).
>
> As a reference for the reviewer, we have personally reached out to the authors of [point-voxel-diffusion](https://arxiv.org/abs/2104.03670) and [point-straight-flow](https://arxiv.org/abs/2212.01747) [7,8] (PVD and PSF in **Table 3**) to inquire about training time. In both cases, the authors indicated training lasted several days (3-4 for PVD and 2-3 for PSF), and required significant compute (4 GPUs for PVD and 8 for PSF). WFM achieves comparable quality with fewer resources.
>
> > I believe there would be scalability issues when the number of particles becomes more than 2000.
>
> The reviewer is correct that WFM suffers as the number of particles increases. Again, due to the transformer and estimate of the OT map. While computational costs do increase with point cloud size, this scaling remains practical for most applications.
>
> We refer the reviewer to the [github]([https://github.com/WassersteinFlowMatching/WassersteinFlowMatching](https://github.com/WassersteinFlowMatching/WassersteinFlowMatching)) where we added an [example](https://github.com/WassersteinFlowMatching/WassersteinFlowMatching/blob/main/tutorials/tutorial_point_cloud_2000_wfm.ipynb) of training WFM to generate point-clouds with **2000** particles. In a modest 15 hours of training, (50,000 training steps and batch size of 32), WFM produces high fidelity point-clouds.

---

> > ### Author Response · Authors · 2024-11-19
> > **Response to Reviewer UUcS:**
> >
> > > Overall, the method involves an inaccurate approximation (e.g. standard OT -> entropic OT) and seems to be computationally demanding (e.g. sinkhorn for every iterations). Moreover, its performance results is not convincing for me, too. I believe the benefit of WFM method and the potential applications should be more carefully carried.
> >
> > We appreciate the reviewer’s observations and address the key points as follows:
> >
> > -  The use of entropic OT as an approximation to standard OT is a common and well-established practice in optimal transport literature due to its computational efficiency.
> > -  We now provide a detailed discussion of how entropic OT adheres to flow-matching theory and the trade-offs between accuracy and efficiency (**Appendix A.3 and B**).
> > -  WFM is no more computationally demanding than contemporary methods. In fact, it achieves comparable performance with fewer computational resources.
> > -  While our results on low-dimensional and uniform point clouds match are on par benchmarks, our key contribution lies in WFM’s ability to generate variably-sized, high-dimensional point clouds. To our knowledge, this is a novel capability with substantial potential for applications in fields like computational geometry and single-cell genomics.
> >
> > >Could the authors provide training times, including times for solving entropic OT, for the point cloud generation tasks? Additionally, a comparison with other models, such as diffusion-based point cloud generation methods, would be insightful.
> >
> > We now discuss training times in **Appendix Section E.2** of our revised manuscript.
> >
> > > Does the optimal velocity field vt that minimizes WFM objective guarantee the transportation of p0 to p1? Could the authors provide the theoretical proofs?
> >
> > We thank the reviewer for pointing out this omission. In **Appendix B**, we now mention exactly how one derives the objective. In the case of BW-FM, the objective indeed guarantees transport. For the general WFM, because the optimization is over point-clouds, there is likely error associated with the optimization, which is quite difficult to make precise. In our revised manuscript (**Appendix A.3**), we discuss existing statistical theorems for quantifying geodesics on the basis of samples.
> >
> > >What is the difference of this algorithm compare to mini-batch OT with Wasserstein distance cost (in point clouds)?
> >
> > It is not entirely clear to us what the reviewer is inquiring about. Could they elaborate on what specific aspects they would like us to compare with mini-batch OT using Wasserstein distance for point clouds? We are happy to expand the discussion and conduct additional experiments before the rebuttal deadline.
> >
> > [1] Davis, Oscar, et al. "Fisher flow matching for generative modeling over discrete data." _arXiv preprint arXiv:2405.14664_ (2024).
> >
> > [2] Stark, Hannes, et al. "Dirichlet flow matching with applications to dna sequence design." _arXiv preprint arXiv:2402.05841_ (2024).
> >
> > [3] Cheng, Chaoran, et al. "Categorical Flow Matching on Statistical Manifolds." _arXiv preprint arXiv:2405.16441_ (2024).
> >
> > [4] Bose, Avishek Joey, et al. "Se (3)-stochastic flow matching for protein backbone generation." _arXiv preprint arXiv:2310.02391_ (2023).
> >
> > [5] Amos, Brandon, et al. "Meta optimal transport." _arXiv preprint arXiv:2206.05262_ (2022).
> >
> > [6] Zhang, Lily, et al. "Set norm and equivariant skip connections: Putting the deep in deep sets."  _International Conference on Machine Learning_. PMLR, 2022.
> >
> > [7] Zhou, Linqi, Yilun Du, and Jiajun Wu. "3d shape generation and completion through point-voxel diffusion." _Proceedings of the IEEE/CVF international conference on computer vision_. 2021.
> >
> > [8] Wu, Lemeng, et al. "Fast point cloud generation with straight flows." _Proceedings of the IEEE/CVF conference on computer vision and pattern recognition_. 2023.

---

> > > ### Comment · Reviewer_UUcS · 2024-11-24
> > >
> > > Thank you to the authors for clarifying the novelty and computational costs of the paper. This work is indeed the first to introduce and discuss the permutationally invariant flow matching algorithm, which I acknowledge as a significant strength. Moreover, now I see that there are computational advantages compared to other recent works.
> > >
> > >
> > > However, I remain concerned about the theoretical soundness of the method and its practical implications. In particular, I find issues with certain aspects of the theory and perceive a potential gap between the theoretical framework and its practical implementation. While these concerns were raised in my previous review, I realize that my questions may have been unclear, and I apologize for any lack of clarity.
> > >
> > > To ensure a more clear discussion, I will first outline my understanding of the paper, followed by a series of specific questions. If I have misunderstood something, please let me know.
> > >
> > > $ $
> > >
> > > **General Statement**: Given $\mu \sim p_0$ and $\nu \sim p_1$, my understanding is that the paper obtains $\mu_t$ using a displacement interpolant to train flow matching (FM). However, I would like to focus on the optimal velocity field $v_t$ we aim to learn. The optimal velocity is the "mean direction," i.e., $v_t(\mu_t) = \mathbb{E}[\mu_1 | \mu_t]$. Consequently, the learned velocity field $v_t$ does not connect the two distributions $\mu \sim p_0$ and $\nu \sim p_1$ via Wasserstein geodesics. Instead, $v_t$ is the mean vector field. (I don't mean that this is the problem.)
> > >
> > > Now, I have several followed questions:
> > >
> > > $ $
> > >
> > > **Q1.**  I am not convinced that WFM generally bridges $p_0$ and $p_1$. For instance, let $p_0 = \\{ \delta_0 \\}$ and $p_1 = \\{ \delta_{-1}, \delta_{1} \\}$. In this case, the optimal vector field $v_t$ satisfies $v_t(\delta_0) = 0$. As a result, the optimal vector field does not establish a connection between $p_0$ and $p_1$.
> > >
> > > Given this, I believe it is important for the authors to clarify the assumptions regarding the collection of probability densities and the regularity conditions required for the optimal vector field $v_t$ to bridge $p_0$ and $p_1$. For instance, in the original FM paper, they discussed the assumptions required for the marginal densities and also some other conditions to be satisfied (e.g. Leibniz rule).
> > >
> > > Additionally, even if we set $p_0,p_1 \subset \mathcal{P}_{2,ac}$, I don't think the bridging property would be guaranteed. Moreover, applying Riemannian FM seems also challenging, as the space is not a finite smooth Riemannian manifold.
> > >
> > > In summary, demonstrating that the "mean vector field" $v_t$ bridges $p_0$ and $p_1$ does not seem trivial. Could the authors provide clarification on this aspect?
> > >
> > > $ $
> > >
> > > **Q2.** I find the statement on line 77—"our aim is to learn vector fields acting on the space of probability distributions and match the optimal transport map, which is the geodesic in Wasserstein space"—to be somewhat misleading. The method learns the mean vector field, not a vector field that satisfies the conditions of optimal transport. Additionally, I have reviewed the original FM and Riemannian FM papers, and they do not claim to "match" between two points in this manner. Could the authors clarify this statement?
> > >
> > > $ $
> > >
> > > **Q3.** I think some continuity or non-vanishing condition will be required for $p_0$ and $p_1$. But, if so, I don't think the point cloud is the case (since it is a sum of delta distributions).

---

> ### Author Response · Authors · 2024-11-24
> **Author Response to Remaining Questions:**
>
> Thank you for engaging with our paper! We are grateful the reviewer found our work conceptually novel and noted its computational efficiency. We thank the reviewer for clearly outlining their remaining theoretical concerns, which now we aim to resolve.
>
> **Response to General Statement**:
>
> Do you have a reference for where the velocity $v_t(\mu_t)=\mathbb{E}[\mu_1|\mu_t]$ comes from? We believe we have a minor correction here: given the conditional velocities and paths $v_t(\cdot | x_1)$ and $p_t(\cdot|x_1)$, the marginal velocity is $v_t(x)=\mathbb{E}_{x_1}[v_t(x|x_1)p_t(x|x_1)]$. This averaging produces a valid path between from source to target $p_1\rightarrow p_0$ (see Theorem 1 and equation 8-9 in [FM paper](https://arxiv.org/pdf/2210.02747) [1], or equation 7 in the [RFM](https://arxiv.org/abs/2302.03660) [2] paper), which as the reviewer points out is **not** the Wasserstein geodesics between $p_1$ and $p_0$.
>
> From here, it may be helpful to clarify that our setting involves two levels of distributions:
>
> 1. The outer level: $\mathfrak{p}_0$ and $\mathfrak{p}_1$ are distributions over the Wasserstein space
>
> 2. The inner level: Random draws $\mu \sim \mathfrak{p}_0$ and $\nu \sim \mathfrak{p}_1$ are themselves distributions (point-clouds or Gaussians)
>
> We learn conditional vector fields between samples by regressing onto the Wasserstein geodesic connecting $\mu \rightarrow \nu$. While the FM objective still holds (see derivation in **Appendix Section B**), producing a map $\mathfrak{p}_0 \rightarrow \mathfrak{p}_1$, we do not attempt to learn the Wasserstein geodesic between $\mathfrak{p}_0$ and $\mathfrak{p}_1$.
>
> We have clarified this nuanced point in our re-revision (blue text in pdf).
>
> **Response to Q1**:
>
> This has been an illuminating example for us to consider, and has led to us realizing the following connection: **flow matching is a special case of WFM when the measures are Diracs.** This is because the OT map between $\mu = \delta_{x_{0}}$ and $\nu = \delta_{x_{1}}$ is the straightforward linear path $x_{1} - x_{0}$, so the WFM objective turns into the standard FM one. Thus, the reviewer's example, while insightful, falls outside both FM assumptions and our generative setting.
>
> Given this, the reviewer is correct that in their described scenario, where the source measure consists of only a single point-cloud ($p_0: \mu=\delta_{0}$ with probability $1$) while target has two ($p_1: \nu = \delta_{-1}$ or $\nu = \delta_{1}$, each with probability $0.5$), there is no map from source to target. So, in the standard FM case, if the source is a single point, there is no deterministic function which splits it into 2 to draw from the target.
>
> This has helped us clarify the following two key assumptions under which WFM operates (**all** of our experimental settings fit into these):
>
> 1.  Outer continuity: the source $\mathfrak{p}_0$ is a continuous measure over distributions (e.g., the spiral of Gaussians in **Figure 2** or shapes of all cars for point-clouds). Similar regularities assumptions for the source distribution are also required in standard FM (see Section 2 in [FM paper](https://arxiv.org/pdf/2210.02747) [1])
>
> 2.  Inner continuity: samples $\mu \sim \mathfrak{p}_0$ and $\nu \sim \mathfrak{p}_1$ are continuous distributions:
>
>
> - Natural when $\mu$ and $\nu$ for Gaussians
>
> - For point-clouds, $\mu$ and $\nu$ are interpreted as continuous shapes where point-clouds are drawn from them.
>
> These assumptions ensure:
>
> 1.  A transport path exists between $\mathfrak{p}_0$ and $\mathfrak{p}_1$
>
> 2.  Deterministic OT maps exist between $\mu \sim \mathfrak{p}_0$ to $\nu \sim \mathfrak{p}_1$
>
> Does it make sense? Do you agree how the example connects WFM and FM, and is outside of the assumptions of both of them?
>
> The reviewer correctly points out that the infinite dimensional Wasserstein space is not a Riemannian manifold. While we make note of this in our manuscript (line 186), we do not fully discuss this matter for brevity. Indeed, the relevant features of Riemannian manifolds for Flow Matching, which are the existence of a tangent space and exponential/logarithmic maps, are still true in Wasserstein space
>
> **Response to Q2**:
>
> We did not mean to imply that we learn the OT map — we agree this would be misleading. Instead, WFM learns to map source to target by regressing onto OT maps (Wasserstein geodesics) between *samples* $\mu \sim p_0$ and $\nu \sim p_1$, which are themselves distributions. We have clarified this in lines 266-268 of the latest revision. Can you please let us know if it reads okay now?
>
> > *“WFM learns to map source to target by regressing onto Wasserstein geodesics between samples $\mu \sim \mathfrak{p}_0$ and $\nu \sim \mathfrak{p}_1$, rather than learning the OT map between $\mathfrak{p}_0$ and $\mathfrak{p}_1$.”*

---

> ### Author Response · Authors · 2024-11-24
> **Author Response to Remaining Questions:**
>
> **Response to Q3**:
>
> While point-clouds are discrete distributions, we interpret them as samples from underlying continuous shapes. This continuity assumption, now explicitly stated in our re-revision, ensures the existence of OT maps between them. Do you agree this justifies the experimental settings?
>
> We again thank the reviewer for their insightful discussion.
>
> [1] Yaron Lipman, Ricky TQ Chen, Heli Ben-Hamu, Maximilian Nickel, and Matt Le. Flow matching for generative modeling. arXiv preprint arXiv:2210.02747, 2022.
>
> [2] Ricky TQ Chen and Yaron Lipman. Riemannian flow matching on general geometries. arXiv preprint arXiv:2302.03660, 2023.

---

> ### Comment · Reviewer_UUcS · 2024-11-27
>
> Thank you for your clarification—I believe the explanations provided for Q2 and Q3 are clear. Additionally, I understood the explanation of FM; FM assumes positive density throughout the domain, and with your inner/outer assumption, this counter-example can be easily eliminated.
>
> Turning back to the paper, while the concept is very novel, I still find both the theoretical and experimental contributions lacking. Regarding the experiments, although the paper does not include many comparisons, it does demonstrate potential applicability. Therefore, this is not my primary concern. However, my primary concern lies with the lack of rigorous theoretical grounding. As with FM or Riemannian FM, I believe a precise theoretical discussion on the matching between $p_0$ and $p_1$ is crucial.
>
> Could the authors include a concrete theorem in the manuscript under the assumption that $p_0$ is a continuous distribution on the Wasserstein space? At present, this is not evident, as we cannot rely on the smoothness assumptions of the manifold or vector field typically used in the Riemannian setting.

---

> ### Author Response · Authors · 2024-12-02
> **Revision with theoretical concerns**
>
> We are grateful to the reviewer for their continued engagement with our work. We are pleased they were satisfied with our explanations for **Q2** and **Q3** and for noting the conceptual novelty and potential applicability of our work.
>
> > Regarding the experiments, although the paper does not include many comparisons...
>
> It is worth mentioning that over the Bures--Wasserstein manifold, there are no other tractable, large-scale procedures available for learning curves of measures on this space. For high-dimensional point-clouds, other methods do not scale when learning generative models due to the fact that they employ voxelization as a key aspect of their algorithms.
>
> The benchmarking in our manuscript follows established standards for point-cloud generative modeling. Following [point-voxel-diffusion](https://arxiv.org/abs/2104.03670) and [point-straight-flow](https://arxiv.org/abs/2212.01747) [1,2] (PVD and PSF in **Table 3**), we also measure the 1-Nearest-Neighbour (1-NN) accuracy for generation of planes, cars and chairs to assess the quality of our model.
>
> In fact, our work goes beyond previous efforts, as we can generate high-dimensional and variable sized-point clouds. No previous work is capable of this, and here too we validate the fidelity of our model using the same 1-NN metric.
>
> > However, my primary concern lies with the lack of rigorous theoretical grounding… cannot rely on the smoothness assumptions of the manifold or vector field typically used in the Riemannian setting.
>
> Regarding your theoretical concerns, we note that the regularity of the metric $g$ and measures $\mathfrak{p}_0,\mathfrak{p}_1 \in \mathcal{P}(\mathcal{M})$ are not rigorously discussed in the original FM or RFM papers. Indeed, Appendix A in [RFM](https://arxiv.org/abs/2302.0366) [3] simply assumes any and all conditions required for their proofs to work; we eagerly make these same assumptions and we now clearly mention this. Their key assumption is the positivity of the probability path, which is covered by our inner- and outer-continuity, as you have agreed.
>
> With this in mind, we included a concrete proposition and corollary in the newly added **APPENDIX B.1** (purple text in our revised manuscript, see [github](https://github.com/WassersteinFlowMatching/WassersteinFlowMatching/blob/main/WassersteinFlowMatching.pdf)), which we briefly describe here:
>
> Recall that our probability paths are of the same form as Eq (7) of the RFM paper, which for $\mu \sim \mathfrak{p}_0$ and $\nu \sim \mathfrak{p}_1$, is
>
> $$ \mu_t = \exp_{\mu}((1-t)\log_{\mu}(\nu)) = ((1-t){\mathrm{id}} + tT^{\mu\to\nu})_\sharp\mu\,.$$
>
> We prove in Proposition B.1 that the squared $2$-Wasserstein distance is a valid pre-metric in the sense described by the RFM paper. From this, we compute explicitly the following (conditional) vector field (following Eq (13) in the RFM paper):
>
> $ {\mathfrak{u}}_t(\mu_t|\nu)  = -\frac{1}{1-t}{\mathrm{grad}}(W_2^2(\cdot, \nu))(\mu_t) = v_t,$
>
> where $v_t$ is the optimal transport vector field that we regress onto. As a corollary (**Corollary B.2** in our revised manuscript), the conditional paths and conditional vector fields are well-defined, and thus results in a valid FM objective function. We note that in all instances in our manuscript, the OT map is well-defined and non-singular, as we assume all measures are densities with respect to Lebesgue measure.
>
> All that said, we again thank the reviewer for their detailed assessment of our work, and hope we have answered any remaining concerns.
>
> [1] Zhou, Linqi, Yilun Du, and Jiajun Wu. "3d shape generation and completion through point-voxel diffusion." _Proceedings of the IEEE/CVF international conference on computer vision_. 2021.
>
> [2] Wu, Lemeng, et al. "Fast point cloud generation with straight flows." _Proceedings of the IEEE/CVF conference on computer vision and pattern recognition_. 2023.
>
> [3] Ricky TQ Chen and Yaron Lipman. Riemannian flow matching on general geometries. arXiv preprint arXiv:2302.03660, 2023.

---

### Official Review · Reviewer_dBX1 · 2024-11-03

**Soundness:** 3
**Presentation:** 4
**Contribution:** 3
**Rating:** 8
**Confidence:** 3

**Summary:**

The paper demonstrates an application of Riemannian Flow Matching for the novel problem of generating probability measures on the Wasserstein manifold, in particular, for the case of the Bures–Wasserstein space - i.e. distributions over Gaussian distributions, and for point-cloud data considered as empirical measures. The authors present tractable training objectives and sampling schemes for both settings, relying on closed-form solutions available in the Bures–Wasserstein space, and the approximation of the optimal transport map by entropic couplings in the point-cloud setting. Notably, their methodology allows for the generation of point-clouds with non-uniform size. They further validate their framework via experiments with single-cell genomics data and 2D and 3D point-clouds, presenting competitive results and improved scalability.

**Strengths:**

The paper tackles a novel and important problem within generative modelling of the generation of probability measures, with particular relevance for single-cell genomics data. The authors present a sound methodology in the particular cases of modelling distributions of Gaussian distributions and point-cloud data, with the notable attractive feature of being able to model point-cloud data of non-uniform sizes which is to the best of my knowledge, the first instance of this being tractable; this is backed up with convincing experimental validation. In addition, the paper is well-written and presented.

**Weaknesses:**

A potential criticism of the work is that the main components of the methodology such as Riemannian Flow Matching, closed-form solutions for geodesics etc. on the Bures–Wasserstein space, the entropic estimation of optimal transport maps etc. have been introduced elsewhere, and the main contribution of the paper is simply the original combination of these elements applied to their problem. In addition, the authors could have further discussed other settings in which generative modelling on the Wassertein manifold could be useful, such as amortising Bayesian inference, beyond the two cases considered in the paper.

**Questions:**

I do not have any questions.

---

> ### Author Response · Authors · 2024-11-19
> **Response to Reviewer dBX1:**
>
> > The paper tackles a novel and important problem within generative modelling of the generation of probability measures, with particular relevance for single-cell genomics data. The authors present a sound methodology in the particular cases of modelling distributions of Gaussian distributions and point-cloud data, with the notable attractive feature of being able to model point-cloud data of non-uniform sizes which is to the best of my knowledge, the first instance of this being tractable; this is backed up with convincing experimental validation. In addition, the paper is well-written and presented.”
>
> We thank the reviewer for their positive assessment of our work. We agree with the reviewer that the ability to model point-clouds data with inhomogeneous sizes is a particularly exciting feature of WFM.
>
> We further stress WFM’s scalability to high-dimensional data through the Transformer architecture. This unlocks key application to single-cell genomics, namely the ability to generate microenvironments from spatial transcriptomics data, which are point-cloud in gene-expression space. This ability to model cellular structure in a generative manner is an entirely novel concept made possible by WFM, and brings to the promise of generative models whose impact in single-cell has been immense (see [Virtual cell](https://arxiv.org/abs/2409.11654) efforts [1]), to the spatial domains.
>
> > A potential criticism of the work is that the main components of the methodology such as Riemannian Flow Matching, closed-form solutions for geodesics etc. on the Bures–Wasserstein space, the entropic estimation of optimal transport maps etc. have been introduced elsewhere, and the main contribution of the paper is simply the original combination of these elements applied to their problem.”
>
> As noted in our response to  **reviewer R81K**, WFM is indeed an instantiation of Riemannian Flow Matching, as are many other works for generative modeling on statistical domains ([Fisher](https://arxiv.org/abs/2405.14664), [Dirichlet](https://arxiv.org/abs/2402.05841), and [Categorical Flow Matching](https://arxiv.org/abs/2405.16441) [2,3,4]).
>
> Our innovation lies in harnessing the Wasserstein manifold, particularly suited for Gaussians (Bures--Wasserstein) and discrete measures (point-clouds). By combining OT map estimation with flow matching via transformers, WFM is, to our knowledge, the first generative model for variable-sized, high-dimensional point-clouds, addressing critical needs in single-cell genomics and 3D modeling.
>
> > In addition, the authors could have further discussed other settings in which generative modelling on the Wassertein manifold could be useful, such as amortising Bayesian inference, beyond the two cases considered in the paper.
>
> Thank you for pointing this out; this is a great suggestion. We did not think of this application while preparing our manuscript, but it would certainly be worthwhile to explore this in the future.
>
> [1] Bunne, Charlotte, et al. "How to build the virtual cell with artificial intelligence: Priorities and opportunities." _arXiv preprint arXiv:2409.11654_ (2024).
>
> [2] Davis, Oscar, et al. "Fisher flow matching for generative modeling over discrete data." _arXiv preprint arXiv:2405.14664_ (2024).
>
> [3] Stark, Hannes, et al. "Dirichlet flow matching with applications to dna sequence design." _arXiv preprint arXiv:2402.05841_ (2024).
>
> [4] Cheng, Chaoran, et al. "Categorical Flow Matching on Statistical Manifolds." _arXiv preprint arXiv:2405.16441_ (2024).

---

> > ### Comment · Reviewer_dBX1 · 2024-11-28
> > **Response to the authors**
> >
> > Thanks to the authors for the response. I will maintain my score.

---

### Official Review · Reviewer_R81K · 2024-11-04

**Soundness:** 2
**Presentation:** 3
**Contribution:** 2
**Rating:** 5
**Confidence:** 3

**Summary:**

This paper proposes a new type of generative model based on flow matching, i.e., Wasserstein flow matching. The proposed model allows for working with the *distributions of distributions* and was tested in a variety of experimental setups.

**Strengths:**

The proposed method can be used to deal with families with distributions, i.e., distributions over distributions. It was tested on a variety of Gaussian-based where it shows better performance than naive baselines.

**Weaknesses:**

My main concern is related to the limited significance of the developed methodology. As the authors themselves admit (ines 235-236), the main contribution of this article is the developed algorithm. From a theoretical point of view, the method is based on the results proposed in previous works (Ambrosio et al., 2008, Lambert et al., 2022). At the same time, the authors note that these theoretical results are correct if the algorithm is applied to absolutely continuous measures, which is not true in the case when the measures are approximated as point clouds. Thus, it is not absolutely fair to apply the algorithm to any measures other than Gaussians.

Meanwhile, the theoretical groundlessness of the algorithm could be overlooked if it showed inspiring results from a practical point of view. However, the method demonstrates good results only when working with distributions that can be approximated by families of Gaussian distributions (including the spatial transcriptomics dataset, see lines 414-420). In the case of working with more general measures (point clouds), the method shows results that do not outperform those of its competitors (see Table 3 and authors comment in lines 95-96).

In addition, the experimental part could be improved by including more baselines, for example in problems from the single cell biology domain. In this regard, you might be interested in considering the methods from two recent papers that also rely on flow matching and consider problems from single cell biology (Klein et al., 2024, Eyring et al., 2024).

**Overall**, in my opinion, the presented practical results alone are not enough to publish the paper at the current conference.

**Questions:**

- The notation $\overline{(\cdot)}^{L^2(\mu)}$ is not sufficiently clear in lines 182-183. Could you please write rigorously what is meant here? And I recommend moving such kind of notations in the corresponding part of Section 2.

**References.**

Luigi Ambrosio, Nicola Gigli, and Giuseppe Savare ́. Gradient flows in metric spaces and in the space of probability measures. Lectures in Mathematics ETH Zu ̈rich. Birkha ̈user Verlag, Basel, second edition, 2008. ISBN 978-3-7643-8721-1.

Marc Lambert, Sinho Chewi, Francis Bach, Silve`re Bonnabel, and Philippe Rigollet. Variational inference via Wasserstein gradient flows. Advances in Neural Information Processing Systems, 35:14434–14447, 2022.

Klein, D., Uscidda, T., Theis, F., \& Cuturi, M. (2024). GENOT: A Neural Optimal Transport Framework for Single Cell Genomics. NeurIPS 2024

Eyring, L., Klein, D., Uscidda, T., Palla, G., Kilbertus, N., Akata, Z., \& Theis, F. J. Unbalancedness in Neural Monge Maps Improves Unpaired Domain Translation. ICLR 2024


**After rebuttal.** In the last response of the authors, they have clarified several theoretical aspects of their work. That is, they establish the conditions guaranteeing the validity of their conditional flows. I appreciate these important additions. However, overall, I still have some concerns regarding this paper, e.g., moderate performance of the approach in low dimensions with an absence of baselines in high dimensions (I found out that here the 1-NN accuracy seems to be also quite moderate ~ 60%), some gaps between theory and practical implementation. I see the potential in this paper, thank the authors for their work during the rebuttal but currently keep my score.

---

> ### Author Response · Authors · 2024-11-19
> **Response to Reviewer R81K:**
>
> **Reviewer  R81K:**
>
> >The proposed method can be used to deal with families with distributions, i.e., distributions over distributions. It was tested on a variety of Gaussian-based where it shows better performance than naive baselines.
>
> We are grateful for this constructive review of our work. We appreciate that the reviewer noted the capability of our method to model families of distributions, which is an essential contribution of WFM. In particular, we aimed to address the challenge of generative modeling in the space of distributions over distributions by leveraging the Wasserstein geometry, allowing our method to capture point-clouds and Gaussians effectively.
>
> > My main concern is related to the limited significance of the developed methodology. As the authors themselves admit (lines 235-236), the main contribution of this article is the developed algorithm.
>
> Our work is indeed an instantiation of Riemannian Flow Matching, which we emphasized throughout the manuscript. We take this moment to point out that many previous successful applications of flow matching to non-trivial domains, such as [Fisher](https://arxiv.org/abs/2405.14664), [Dirichlet](https://arxiv.org/abs/2402.05841), [Categorical](https://arxiv.org/abs/2405.16441) and [SE(3) Flow Matching](https://arxiv.org/abs/2310.02391) [1,2,3,4] are also specific instantiations of Riemannian Flow Matching.
>
> We stress three novelties of our work:
> 1.  We formally lift the flow matching problem to the Wasserstein manifold, enabling unified treatment of both Gaussians and point-clouds, both cases being novel application domains for spatial transcriptomics.
> 2.  We combine the transformer architecture with entropic OT maps for learnable point-cloud flows
> 3.  Combining (1) + (2), we have the unique ability to generate high-dimensional and variable-sized point-clouds, a capability not found in the existing literature (to the best of our knowledge).
>
> > From a theoretical point of view, the method is based on the results proposed in previous works (Ambrosio et al., 2008, Lambert et al., 2022). At the same time, the authors note that these theoretical results are correct if the algorithm is applied to absolutely continuous measures, which is not true in the case when the measures are approximated as point clouds. Thus, it is not absolutely fair to apply the algorithm to any measures other than Gaussians.
>
>
> We understand the concerns of the reviewer, however, it is important to note some key aspects:
> -   The space of Gaussians is effectively the only non-trivial space of probability measures for which we have closed-form geodesics.
> -   Point-clouds can be readily interpreted as samples from a population distribution (consider a shape as a continuous distribution, and point-clouds constitute samples from said distribution). This is the more common regime where OT is applicable in practice. Here, practitioners who are interested in the OT cost are forced to estimate it, often using entropic optimal transport, as introduced by [Cuturi (2013)](https://arxiv.org/abs/1306.0895) [5]. This has received immense attention in the literature, and recently, [Pooladian and Niles-Weed (2021)](https://arxiv.org/abs/2109.12004) [6] proposed an estimator for the OT map based on entropic optimal transport, which has found its way to the widely used open-source toolbox [OTT-JAX](https://ott-jax.readthedocs.io/en/latest/).
>
> To this end, we refer the reviewer to the newly written **Appendix A.3**, where we discuss the delicate point of estimating geodesics using entropic optimal transport.
>
> > Meanwhile, the theoretical groundlessness of the algorithm could be overlooked if it showed inspiring results from a practical point of view. However, the method demonstrates good results only when working with distributions that can be approximated by families of Gaussian distributions (including the spatial transcriptomics dataset, see lines 414-420). In the case of working with more general measures (point clouds), the method shows results that do not outperform those of its competitors (see Table 3 and authors comment in lines 95-96).
>
> For the **low-dimensional and uniformly sized** point-cloud experiments, we agree that our algorithm does not achieve state-of-the-art results. However, as shown in **Table 2**, the use of transformers and the (entropic) OT map makes WFM the first method (to our knowledge) we can generate **non-homogenous and high dimensional point-clouds**; please see our general response.
>
> Finally, as we note in our response to **reviewer UUcS**, WFM requires significantly less resources to train compared to other approaches. Per our personal correspondence with the authors of these papers, [point-voxel-diffusion](https://arxiv.org/abs/2104.03670) and [point-straight-flow](https://arxiv.org/abs/2212.01747) [7,8] (PVD and PSF in **Table 3**), require 4-8 GPUs for several days. In stark contrast, our method needs only a single GPU for 3 days to produce competitive results.

---

> ### Author Response · Authors · 2024-11-19
> **Response to Reviewer R81K:**
>
> > In addition, the experimental part could be improved by including more baselines, for example in problems from the single cell biology domain. In this regard, you might be interested in considering the methods from two recent papers that also rely on flow matching and consider problems from single cell biology (Klein et al., 2024, Eyring et al., 2024).”
>
> We are familiar with both of these works, however, to our knowledge, their aims are different from ours. Specifically, [Klein et al. (2024)](https://arxiv.org/abs/2310.09254) and [Eyring et al. (2024)](https://arxiv.org/abs/2311.15100) [9, 10] do not explore flow matching on the Wasserstein space; rather, they focus on flows between samples which are individual data points rather than point clouds. In contrast, WFM operates on the space of distributions, with each sample representing a complete point-cloud. These are incomparable tasks, though we have added a detailed discussion in **Section 2.2: Related Work**.
>
> Additionally, WFM introduces an entirely novel concept of generating cellular environments—specifically, point-clouds derived from spatial transcriptomics data. This capability is highly relevant, as it provides unprecedented insight into spatial microenvironments, a critical aspect of cellular biology that no previous work has addressed in this way.
>
> > The notation $\overline{(\cdot)}^{L^2(\mu)}$ is not sufficiently clear in lines 182-183. Could you please write rigorously what is meant here? And I recommend moving such kind of notations in the corresponding part of Section 2.
>
> The closure of a set is the set and all its cluster points with respect to a topology (you can think of it as the boundary of a set). This is a minor technical point required in the definition of the tangent space, and we included a brief sentence in the main text to aid the reader.
>
> [1] Davis, Oscar, et al. "Fisher flow matching for generative modeling over discrete data." _arXiv preprint arXiv:2405.14664_ (2024).
>
> [2] Stark, Hannes, et al. "Dirichlet flow matching with applications to dna sequence design." _arXiv preprint arXiv:2402.05841_ (2024).
>
> [3] Cheng, Chaoran, et al. "Categorical Flow Matching on Statistical Manifolds." _arXiv preprint arXiv:2405.16441_ (2024).
>
> [4] Bose, Avishek Joey, et al. "Se (3)-stochastic flow matching for protein backbone generation." _arXiv preprint arXiv:2310.02391_ (2023).
>
> [5] Cuturi, Marco. "Sinkhorn distances: Lightspeed computation of optimal transport." _Advances in neural information processing systems_ 26 (2013).
>
> [6] Pooladian, Aram-Alexandre, and Jonathan Niles-Weed. "Entropic estimation of optimal transport maps." _arXiv preprint arXiv:2109.12004_ (2021).
>
> [7] Zhou, Linqi, Yilun Du, and Jiajun Wu. "3d shape generation and completion through point-voxel diffusion." _Proceedings of the IEEE/CVF international conference on computer vision_. 2021.
>
> [8] Wu, Lemeng, et al. "Fast point cloud generation with straight flows." _Proceedings of the IEEE/CVF conference on computer vision and pattern recognition_. 2023.
>
> [9] Klein, Dominik, et al. "Generative Entropic Neural Optimal Transport To Map Within and Across Spaces." _arXiv preprint arXiv:2310.09254_ (2023).
>
> [10] Eyring, Luca, et al. "Unbalancedness in Neural Monge Maps Improves Unpaired Domain Translation." _arXiv preprint arXiv:2311.15100_ (2023).

---

> > ### Comment · Reviewer_R81K · 2024-12-02
> > **Official comment**
> >
> > Dear authors,
> >
> > thank you for additional clarifications. However, after skimming through your discussion with other reviewers (especially, the reviewer UUcS), I have become more sure in my concerns regarding the theoretical groundlessness of the proposed algorithm.  For example, it became visible that, in general, the established algorithm is not guaranteed to learn the OT map between the input and target families of measures. It is an important point since you suggest to use the method in tasks from single-cell biology where it is important to understand what your method actually learns. From the practical side, I agree that the fact that your method can generate non-homogenous and high dimensional point-clouds is indeed interesting (Table 2). But it seems to be not an enough contribution by its own.
> >
> > Overall, I still think that the paper needs to be improved from the theoretical and/or experimental part. However, since the authors clarified some of my concerns and made several important changes in the revised version of the paper, I update my score to 5.

---

> > > ### Author Response · Authors · 2024-12-03
> > > **Response to Reviewer concerns:**
> > >
> > > Thank you for raising your score given our initial response!
> > >
> > >
> > > To address theoretical concerns raised by yourself and reviewer UUcS, we have added **Appendix B.1** (purple text in our revised manuscript, see [github]([https://github.com/WassersteinFlowMatching/WassersteinFlowMatching/blob/main/WassersteinFlowMatching.pdf](https://github.com/WassersteinFlowMatching/WassersteinFlowMatching/blob/main/WassersteinFlowMatching.pdf))), and had an extensive discussion with reviewer UUcS, which we encourage you to look it. In summary:
> > >
> > > 1.  We establish that our conditional vector field, derived via the optimal transport map between samples, is well-defined.
> > >
> > > 2.  Show that the squared 2-Wasserstein is a valid pre-metric (**Proposition B.1**).
> > >
> > > 3.  Proving our optimal transport vector fields generate valid conditional probability flows (**Corollary B.2**).
> > >
> > >
> > > As a result, WFM inherits the theoretical guarantees from [Riemannian Flow Matching]([https://arxiv.org/abs/2302.03660](https://arxiv.org/abs/2302.03660)) [1], under the same assumptions used in their original framework.
> > >
> > >
> > > > For example, it became visible that, in general, the established algorithm is not guaranteed to learn the OT map between the input and target families of measures.
> > >
> > > This appears to be a misunderstanding of our method's purpose. The goal of WFM is to learn conditional flows between source and target families of measures, by regressing onto optimal transport vector fields between random distributions $\mu \sim \mathfrak{p}_0$ and $\nu \sim \mathfrak{p}_1$, where $\mathfrak{p}_0$ and $\mathfrak{p}_1$ are measures over the space of distributions (e.g., $\mu$ and $\nu$ are point-clouds or Gaussians). Our WFM algorithm **does not** attempt, nor we do claim it does, to learn the **optimal transport map** between the families of distributions $\mathfrak{p}_0$ and $\mathfrak{p}_1$.
> > >
> > > As we show in the newly added **Appendix B.1**, WFM fully adheres to the framework laid out by Riemannian Flow Matching, inheriting its theoretical guarantees as a **conditional flow matching** procedure, and it learns valid conditional probability paths. Like other Flow Matching methods, we do not attempt to learn the optimal transport map between $\mathfrak{p}_0$ and $\mathfrak{p}_1$ - rather, we learn a valid flow between them.
> > >
> > > > It is an important point since you suggest to use the method in tasks from single-cell biology where it is important to understand what your method actually learns.
> > >
> > > In single-cell genomics, we use WFM to generate cellular microenvironments - point clouds in gene-expression space - from spatial transcriptomics data. The method learns to transform source, noisy, point-clouds into biologically meaningful microenvironments, precisely matching the distribution of cellular neighborhoods observed in real tissue. This is a novel application in the field, made possible by WFM's unique ability to handle high-dimensional, variable-sized point clouds.
> > >
> > > Thank you for continuing to engage with our paper. We hope to have cleared up any remaining concerns.
> > >
> > > [1] Ricky TQ Chen and Yaron Lipman. Riemannian flow matching on general geometries. arXiv preprint arXiv:2302.03660, 2023.

---

> > > > ### Author Response · Authors · 2024-12-04
> > > > **New Appendix B.1 (Part 1)**
> > > >
> > > > To ease the reviewing process, we thought to post our short new appendix here for your convenience. Thank you for continuing to engage with our paper
> > > >
> > > > ---
> > > > ## Validity of Conditional Flows
> > > >
> > > > The key idea behind Flow Matching is that the (tractable) conditional velocity probability paths come together to form correct marginal velocities and paths from source distribution to target. In Riemannian Flow Matching, given a source and target $\mathfrak p_1, \mathfrak p_0 \in \mathcal P (\mathcal M)$ distributions over *Riemannian* manifold $\mathcal{M}$, the conditional velocity field from $\mathfrak p_0$ to a sample $\nu\sim\mathfrak p_1$ is generated by flowing from each point $\mu\sim\mathfrak p_0$ towards $\nu$ using the geodesic path between them. This produces the conditional probability path $\mathfrak p_{t}(\cdot|\nu)$ generated by the vector field $\mathfrak u_{t}(\cdot|\nu)$, which obey the continuity equation for *Riemannian* manifolds:
> > > >
> > > > $$   \partial_t\mathfrak p_t(\cdot|\nu) + \text{div}_{\mathcal M}(\mathfrak p_t(\cdot|\nu)\mathfrak u_t(\cdot|\nu)) = 0 $$
> > > >
> > > > where $\text{div}_{\mathcal{M}}$ is the Riemannian divergence operator. Following theorem 1 in Lipman et al. (2022) and equation 7 in Chen and Lipman (2024), the marginal vector field:
> > > >
> > > > $$\mathfrak u_t(\mu_t) = \int_{\mathcal{M}} \mathfrak u_t(\mu_t|\nu)\frac{\mathfrak p_t(\mu_t|\nu)}{\mathfrak p_t(\mu_t)} {\mathrm{d}} \mathfrak p_{1}(\nu)$$
> > > >
> > > > generates the marginal probability path:
> > > >
> > > > $$\mathfrak p_t(\mu_t) =\int_{\mathcal{M}} \mathfrak p_t(\mu_t|\nu) {\mathrm{d}} \mathfrak p_{1}(\nu)$$
> > > > as $\mathfrak p_t(\mu_t)$ and $\mathfrak u_t(\mu_t)$ together obey the continuity equation. The proof requires regularity constraints on $\mathfrak p_t(\mu_t|\nu)$ and $\mathfrak u_t(\mu_t|\nu)$, which we assume as in Chen and Lipman (2024). A withstanding requirement that we need is that a unique optimal transport map exists between *any* two measures $(\mu,\nu) \in \mathfrak p_0 \times \mathfrak p_1$. This can be ensured if, for example, all measures in $\mathfrak p_0$ have a density, which is assumed throughout the text. Positivity of the path $\mu_t$ can be ensured through a myriad of conditions (e.g., source and target measures have positive density over the same support). In the explicit Gaussian-to-Gaussian case (i.e., the Bures--Wasserstein manifold), this assumption is trivially satisfied as the manifold is a *genuine* finite-dimensional Riemannian manifold and the Riemannian machinery of Chen and Lipman (2024) goes through with no modifications.
> > > >
> > > > ## Conditional flows and paths for WFM
> > > > To continue, we require the following proposition. Recall that a pre-metric is a function ${\mathrm{d}}:\cal M \times \cal M \to \mathbb R$ that satisfies the following three requirements
> > > > 1. ${\mathrm{d}}(x,y) \geq 0$ for all $x,y \in \cal M$,
> > > > 2. ${\mathrm{d}}(x,y) = 0$ if and only if $x=y$,
> > > > 3. $\nabla_x {\mathrm{d}}(x,y) \neq 0$ if and only if $x\neq y$.
> > > >
> > > > **Proposition B.1** The $2$-Wasserstein distance defines a valid pre-metric over $\cal P_{2,\rm{ac}}(\mathbb R^d)$.
> > > >
> > > > **Proof:** As the $2$-Wasserstein distance is a metric (Villani (2008), Santambrogio (2015)), the first and second conditions of a pre-metric are trivially satisfied. For the third condition, we note that for $\mu,\nu \in \cal P_{2,\rm{ac}}(\mathbb R^d)$
> > > > \begin{align}
> > > >     \nabla \tfrac12 W_2^2(\cdot,\nu)(\mu) = -\log_{\mu}(\nu) = -2(T^{\mu\to\nu} - \rm{id})
> > > > \end{align}
> > > > For a proof of this computation, see Altschuler et al (2021) or Zemel and Panaretos (2019). By chain-rule, we have that
> > > > \begin{align*}
> > > >     \nabla \tfrac12 W_2^2(\cdot,\nu)(\mu) = W_2(\mu,\nu)\nabla W_2(\cdot,\nu)(\mu) = -\log_{\mu}(\nu)
> > > > \end{align*}
> > > > This latter quantity is only zero if and only if $\mu=\nu$, and thus the proof is complete.

---

> ### Author Response · Authors · 2024-12-04
> **New Appendix B.1 (Part 2)**
>
> We now compute the conditional vector field as described by Eq. (13) by Chen and Lipman (2024) as a function of our choice of pre-metric ${\rm{d}}$, which we recall here, is a conditional vector field written as
> $$\mathfrak u_t(x_t|y) = (\tfrac{\rm{d}}{\rm{d} t}\ln(1-t)) {\rm{d}}(x_t,y) \nabla_x {\rm{d}}(x_t,y)/\|\nabla_x {\rm{d}}(x_t,y)\|^2_{g(x_t)} \quad \quad (23)$$
>
> where $g$ is the metric on the tangent space at $x_t$, and $x_t$ is a geodesic in the form
>
>   $$x_t = \exp_{x_0}((1-t)\log_{x_0}(x_1))\,.$$
>
> Note that our geodesics are precisely of the form
>
> $$ \mu_t = \exp_\mu((1-t)\log_\mu(\nu))\,$$
>
>
> Where we recall $\exp_{\mu}(v) = (\mathrm{id}+v)_{\sharp}\mu$ and $\log\mu(\nu) = T^{\mu \to \nu} - \mathrm{id}$.
>
> Replacing the general pre-metric with the $2$-Wasserstein distance in (23), as well as the metric over the tangent space $\cal T_{\mu_t}\cal P_{2,\rm{ac}}(\mathbb R^d)$ (which is now $L^2(\mu_t)$), we compute
> $$
> \begin{split}
>     \mathfrak u_t(\mu_t|\nu) &= (\tfrac{\rm{d}}{\rm{d} t}\ln(1-t)) W_2^2(\mu_t,\nu) \nabla W_2^2(\mu_t,\nu)/\|\nabla W_2^2(\mu_t,\nu)\|^2_{L^2(\mu_t)} \\
>     &= \frac{1}{1-t} (T^{\mu_t \to \nu} - {\rm{id}}) \\
>     &= \frac{-1}{1-t}(T^{\mu\to\nu} \circ (T_t)^{-1} - {\rm{id}})\\
>     &= \frac{1}{1-t}(T^{\mu\to\nu} - T_t )\circ (T_t)^{-1}\,
> \end{split}
> $$
> where we make use of the fact that $T^{\mu_t\to\nu} = T^{\mu\to\nu}\circ (T^{\mu_t\to\nu})^{-1}$ (see e.g., Section 2 of Carlen and Gangbo (2003)), and that $T_t = (1-t){\rm{id}} + t{T^{\mu\to\nu}}$. Simplifying, we obtain
>   $$ \mathfrak u_t(\mu_t|\nu) = \frac{1}{1-t} (T^{\mu\to\nu} - (1-t){\rm{id}} - t T^{\mu\to\nu} )\circ (T_t)^{-1}  = (T^{\mu\to\nu} - {\rm{id}} )\circ (T_t)^{-1} \quad (24)\,$$
> which is precisely the optimal transport vector field $v_t$ stated in (7). Together, we arrive at the following corollary, which validates the probability paths generated by our approach.
>
> **Corollary B.2** The conditional flow $\mathfrak p_t(\cdot|\nu)$ generated by the conditional vector fields defined by (24) satisfy $\mathfrak p_1(\cdot|\nu) = \nu$.
>
> **Proof** This follows immediately from Theorem 3.1 of Chen and Lipman (2024) by passing through the pre-metric construction above.

---

### Author Response · Authors · 2024-11-19
**General Response:**

First, we would like to extend our appreciation to all reviewers for taking the time to review our paper, with many positive comments, notably from **reviewers dBX1 and UUcS**.

To this end, our contributions are concretely:

1.  Conceptual innovation of extending the flow matching framework and lifting it to the space of probability distributions

2.  Strong empirical performance of Bures--Wasserstein Flow Matching for Guassian distributions compared to benchmarks.

3.  Being the first work to perform generative modeling for non-uniform point-clouds.

We take this moment to emphasize that Gaussian measures have recently been demonstrated to model fine-grain cell states and spatial microenvironments in single-cell genomics. Our approach (BW-FM) enables generation of data in this space, which is a technical and computational novelty, where it is not clear how to use standard generative models.

We now address common themes between the reviews of **R81K** and **UUcS**, which were in reference to (1) the use of entropic OT being used as a proxy for OT, and (2) computational and theoretical concerns as a result:

Gaussians are among the few scenarios where closed-form optimal transport maps exist. In most cases, when practitioners want the OT map, they must be estimated from samples, which also exist as known as point-clouds. The entropic OT map was introduced to the machine learning and statistics community as a principled estimator of the OT map on the basis of samples [Pooladian & Niles-Weed, (2021)](https://arxiv.org/abs/2109.12004) [1]. In the wide literature on OT maps, this is one which is both **computationally friendly** that still comes with statistical guarantees.

On the computational front: because OTT-JAX has a GPU-friendly implementation of entropic OT map estimation, the training bottleneck is minimal; requiring only a single GPU while contemporary methods need 4 to 8. Indeed, for uniform and low dimensional settings WFM uses significantly less resources while achieving comparable performance, while being the **only method which generates non-homogenous and high-dimensional samples**. We discuss existing statistical estimation results for the entropic OT map in **Appendix A.3**.

We have revised our manuscript, with changes made in orange text. Specifically, we added an additional discussion paragraph in **Section 2.2**, and added new sections **A.3** and **B** to the appendix.

We are looking forward to a fruitful discussion.

[1] Pooladian, Aram-Alexandre, and Jonathan Niles-Weed. "Entropic estimation of optimal transport maps." _arXiv preprint arXiv:2109.12004_ (2021).

---

### Meta-Review · Area_Chair_ou5y · 2024-12-20

**Metareview:**

In the paper, the authors proposed Wasserstein flow matching, a type of generative model based on flow matching. While all the reviewers agree that the proposed method is novel, the paper still has some major weaknesses, including (1) the gap between theoretical justification and practical implementation of the method; (2) the limited significance of the method, i.e., the experimental results are not thorough. For instance, as pointed out by the reviewer, the experiments with point clouds in high dimensional settings are lacking; (3) the assumptions are not consistent as pointed out by the reviewers.

I myself also read the paper and agree with the reviewers about the current weaknesses of the paper. Therefore, I recommend rejecting the paper at the current stage. The authors are encouraged to revise the paper based on the suggestions of the reviewers for future resubmission.

**Additional Comments On Reviewer Discussion:**

Please refer to the meta-review.

---

### Decision · Program_Chairs · 2025-01-22

Reject